**REPORT**

# Dynein and dynactin move long-range but are delivered separately to the axon tip

Alexander D. Fellows[1], Michaela Bruntraeger[2], Thomas Burgold[2], Andrew R. Bassett[2], and Andrew P. Carter[1]

**Axonal transport is essential for neuronal survival. This is driven by microtubule motors including dynein, which transports cargo from the axon tip back to the cell body. This function requires its cofactor dynactin and regulators LIS1 and NDEL1. Due to difficulties imaging dynein at a single-molecule level, it is unclear how this motor and its regulators coordinate transport along the length of the axon. Here, we use a neuron-inducible human stem cell line (NGN2-OPTi-OX) to endogenously tag dynein components and visualize them at a near-single molecule regime. In the retrograde direction, we find that dynein and dynactin can move the entire length of the axon (>500 μm). Furthermore, LIS1 and NDEL1 also undergo long-distance movement, despite being mainly implicated with the initiation of dynein transport. Intriguingly, in the anterograde direction, dynein/LIS1 moves faster than dynactin/NDEL1, consistent with transport on different cargos. Therefore, neurons ensure efficient transport by holding dynein/dynactin on cargos over long distances but keeping them separate until required.**

## Introduction

The axon relies on microtubule motors and associated proteins to maintain neuronal function. These factors transport cellular components such as RNAs, proteins, organelles, and neurotrophic signals (Maday et al., 2014). Impairment of these transport mechanisms is detrimental to the neuron with mutations and deficits linked to a range of neurological diseases (Sleigh et al., 2019). Due to the organization of axonal microtubules, kinesin motors drive cargos toward the distal tip (the anterograde direction), whereas a single dynein (cytoplasmic dynein-1, hereafter dynein) transports them back to the cell body (the retrograde direction) (Maday et al., 2014). Dynein relies on its cofactor dynactin, cargo-specific activating adaptors, and associated regulators such as LIS1 and NDEL1 to form a motile complex (Reck-Peterson et al., 2018). However, questions such as how far dynein and its cofactors move and whether they do so together remain unanswered.

Dynein is expressed at high levels in neurons (Twelvetrees et al., 2016) making single molecules of motile dynein difficult to visualize above the background of freely diffusing motor. Studies of neurons from a mouse expressing GFP-tagged dynein intermediate chain used local photobleaching to observe movements of up to 15 μm (Ha et al., 2008; Twelvetrees et al., 2016). In contrast, a recent study in HeLa cells used highly inclined and laminated optical sheet (HILO) imaging to visualize single molecules of GFP-tagged dynein heavy chain. This suggested dynein

has a short residence time on microtubules and only undergoes short-range (1–2 μm) movements (Tirumala et al., 2024), leading to the conclusion that long-range transport is achieved by a constant exchange of motile dynein complexes. These studies raise the question of how far dynein motors move in the axon. Can a single motor travel the whole distance from the axon tip back to the cell body, or do cargos continuously replenish their pool of dyneins?

To ask how dynein drives long-range transport in neurons, we used human stem cells that can be differentiated into excitatory cortical neurons (Pawlowski et al., 2017). This enabled us to endogenously tag dynein and its associated proteins, avoiding any artifacts of overexpression (Watson et al., 2023). We used HILO imaging (Ananthanarayanan et al., 2013; Tirumala et al., 2024) combined with SNAP-tag and HaloTag-linked fluorophores to image dynein molecules live in human neurons at a near-single molecule regime. We discovered that both dynein and dynactin are highly processive in the axon. Furthermore, LIS1 and NDEL1, which are thought to play a role in the initiation of dynein transport, also move long distances. Unexpectedly, when analyzing the anterograde transport of dynein and dynactin, we found that they often move separately toward the distal axon. Taken together, our study allows us to better understand how the dynein machinery drives long-range transport in the axon.

[1]Division of Structural Studies, Medical Research Council Laboratory of Molecular Biology, Cambridge, UK;   [2]Wellcome Sanger Institute, Wellcome Genome Campus, Hinxton, UK.

Correspondence to Andrew P. Carter: cartera@mrc-lmb.cam.ac.uk.

# Results and discussion

## iNeurons as a model to study axonal transport

To study axonal transport, we used engineered human embryonic stem cells (hESC) and human induced pluripotent stem cells (hiPSC). These contain a doxycycline-inducible, neurogenin 2 (NGN2) expression cassette in the adeno-associated-virus integration site 1 safe harbor locus (Pawlowski et al., 2017). Upon treatment with doxycycline, the stem cells undergo rapid, homogenous, and highly reproducible differentiation into excitatory cortical neurons (iNeurons) (Hulme et al., 2022; Schörnig et al., 2021; Zhang et al., 2013). In agreement with previous reports of NGN2-driven differentiation, cells begin to display clear neuronal morphology 7 days post induction (DPI) (Fig. S1 A) (Boecker et al., 2020; Pawlowski et al., 2017; Wang et al., 2017; Zhang et al., 2013). By 21–23 DPI, cells have clearly defined axons and dendrites as shown by immunofluorescence staining by SMI-31 and MAP-2, respectively (Fig. S1 B).

To assess axonal transport, cells were plated into microfluidic devices at 2 DPI and cultured until 21–23 DPI (Park et al., 2006) (Fig. 1, A and B). By this time point, axons have grown through the microfluidic grooves into the axonal compartment and are isolated from dendrites. We treated the axonal compartment with organelle-specific markers to label endosomes (Cholera toxin subunit B), lysosomes (lysotracker), and mitochondria (mitotracker) and characterized their movement (Fig. 1 C). Endosomes displayed faster speeds than the other organelles, in agreement with previous observations in mouse primary neuron cultures (Fellows et al., 2020) and another NGN2-induced neuronal model (Boecker et al., 2020) (Fig. 1 D). Also in agreement with previous work, endosomes and lysosomes moved primarily in the retrograde direction, whereas mitochondria displayed a high degree of bidirectional movement (Boecker et al., 2020; Kulkarni et al., 2022) (Fig. 1 E). Given the similarities between our iNeurons and other reported models, we believe they represent an excellent system to study the role of dynein in long-range transport.

## Visualizing single dynein and dynactin molecules in iNeurons

To visualize dynein, we used CRISPR to endogenously tag the N-terminus of the dynein heavy chain with a HaloTag (Los et al., 2008) in our hESCs line (Halo-DYNC1H1, hESCs, homozygous) and differentiated them into iNeurons. We labeled the HaloTag with Janelia fluor extra dyes (JFX 554 or JFX 650), which are brighter and more photostable than the GFP used previously (Banaz et al., 2019; Twelvetrees et al., 2016). We first treated our Halo-DYNC1H1 iNeurons with 1 nM JFX 554 in the axonal compartment to label a subset of molecules for single-molecule imaging (Broadbent et al., 2023). We saw many distinct dynein spots, most of which were freely diffusing (Fig. S1 C and Video 1). We observed rare instances of processive movement (Fig. S1 C and Video 1), suggesting only a subset of dyneins are actively involved in fast axonal transport.

To ask if labeled dyneins are isolated molecules or clusters, we performed a photobleaching analysis (Fig. 2, A–C and Video 2). This proved difficult due to the movement of the dynein spots in and out of the focal plane. Therefore, we treated iNeurons with N-ethyl maleimide (NEM), which traps motors on microtubules (Pfister et al., 1989; Scott et al., 2011) (Fig. 2, B and C). The immobilized dynein spots displayed between 1 and 7 clear photobleaching steps, with 2 steps being the most common (Fig. 2 D and Fig. S1, D and E). We repeated this analysis on iNeurons containing the endogenously tagged ARP11 subunit of dynactin (Halo-ACTR10, hiPSCs, homozygous). We again saw between 1 and 7 photobleaching steps, but now with 1 step being the most frequent (Fig. 2 E). This difference in the distribution of steps likely correlates with the fact there are two copies of the dynein heavy chain in a dynein motor, but only a single ARP11 per dynactin. We also quantified the intensity of each step for both the dynein and dynactin photobleaching and found them to be very similar (dynein: 25.96 ± 4.38 AU, dynactin: 26.82 ± 6.08 AU, Fig. 2 F), suggesting they correspond to bleaching of individual fluorophores. Our data imply that under these imaging conditions, we are capable of detecting single molecules.

## Dynein moves long-range

To address how far dyneins move, we treated the axonal compartment of Halo-DYNC1H1 iNeurons with 200 nM JFX 554/650. This concentration of dye labels dynein close to saturation ensuring as many dynein molecules were labeled as possible. After 20 min, we collected movies in the microfluidic grooves near the somatodendritic compartment using the same imaging conditions as before. Due to the fluidic isolation, any observed fluorescent signal must travel down the axon. A control iNeuron line without any integrated HaloTag showed no fluorescence at this time point (Fig. S1 F). In contrast, when imaging Halo-DYNC1H1 iNeurons, we saw multiple highly processive spots moving in the retrograde direction (Fig. 3 A, Fig. S1 F, and Video 3). The speed of dynein ranged from 0.3 to 5.0 µm/s with an average of 1.76 ± 0.12 µm/s (Fig. 3 B), which agrees with speeds of retrograde organelles in these neurons (Fig. 1 D) and in the literature (Boecker et al., 2020). To assess the level of dynein present in these moving spots we measured the distribution in intensities in a single 30-ms frame and compared them with the intensities of the static NEM-treated spots, described above (Fig. S1 G). The distribution of intensities of NEM-treated spots was between 0 and 200 AU above the background, consistent with our photobleaching data, suggesting one to seven fluorophores with an average intensity of ~26 AU (Fig. 2 F). The moving dyneins showed a narrower distribution between 0 and 100 AU consistent with between 1 and 4 fluorophores present per spot.

We observed run lengths of individual dynein spots of up to 110 µm, approximately the width of the imaging window. The average run length was shorter at 35.19 ± 0.66 µm, although this appears to be limited by the labeled spots going in and out of focus. The use of an endogenous HaloTag on dynein thus allows us to image much longer runs than observed previously (Twelvetrees et al., 2016; Tirumala et al., 2024). However, with this setup, we were unable to conclusively determine if the dyneins we observed had traveled the whole length of the axon.

To address this, we repeated the experiment but imaged it immediately after treatment with the halo dye. If a dynein molecule is stably attached to a cargo, given the average speed observed (~1.76 µm/s, Fig. 3 B), we would expect the first fluorescent dynein molecule to take under 5 min to traverse the

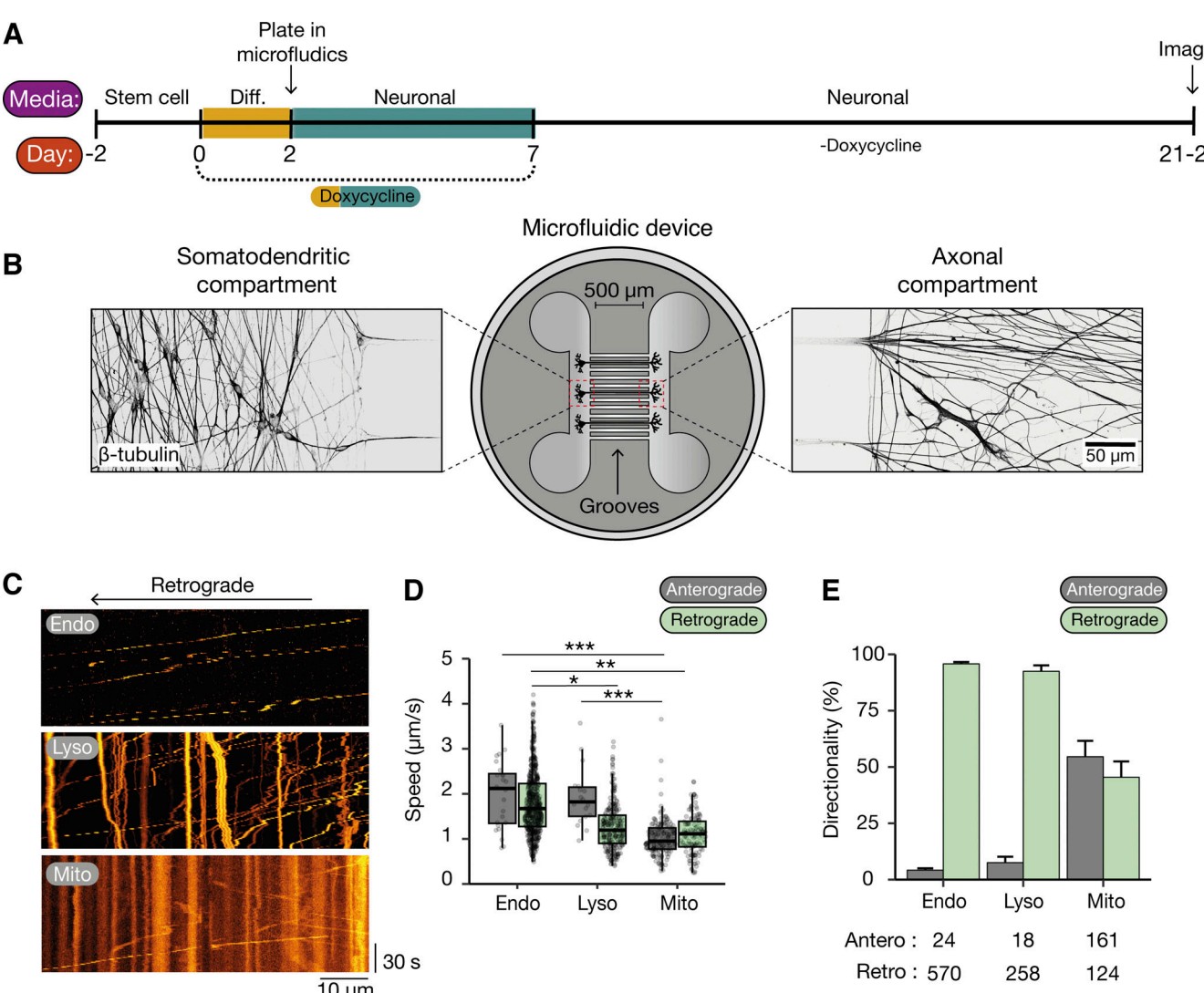

Figure 1. **iNeurons as a model to study dynein-mediated transport. (A)** Schematic of differentiation of NGN2 hESCs/hiPSCs into iNeurons. hESC/hiPSCs were split and 300,000 cells were plated. 2 d later, differentiation media (Diff.) was added to contain doxycycline. At 2 DPI, cells were split again and plated into microfluidics. At 7 DPI, doxycycline was removed from the media and cells were allowed to grow until 21–23 DPI. **(B)** Example image of 21–23 DPI iNeurons in microfluidic device. Cells were fixed and stained with an antibody against β-tubulin. **(C)** Kymographs of endosomes (Endo, CTB AlexaFluor 488), lysosomes (Lyso, Lysotracker Deep Red), and mitochondria (Mito, mitotracker Deep Red FM) in iNeurons at 21–23 DPI. **(D)** The mean speed of endosomes, lysosomes, and mitochondria in both anterograde (gray) and retrograde (light green) directions (Retrograde: endo versus lyso *P = 0.027, endo versus mito **P = 0.0025, lyso versus mito P = 0.71; Anterograde: endo versus lyso P = 0.99, endo versus mito ***P = 0.0002, lyso versus mito ***P = 0.00027), Kruskal–Wallis, Dunn post hoc test, N = 3, Boxplot shows median, first, and third quartiles. Upper/lower whiskers extend to 1.5× the interquartile range. **(E)** The directionality of endosomes, lysosomes, and mitochondria movements in iNeurons at 21–23 DPI (Endosomes: 594 cargoes, 21 videos, N = 3; lysosomes: 276 cargoes, 11 videos, N = 3; mitochondria: 285 cargoes, 22 videos, N = 3). Error bars represent the standard error of the mean (SEM).

500 μm microfluidic groove. Alternatively, if dynein was exchanged on cargo as previously reported (Tirumala et al., 2024), we expected a much slower arrival time of the first signal. This is based on the assumption that once dynein detaches, it loses its ability to move processively and is replaced by "dark" dynein molecules present further along the axon. The labeled dynein will therefore take time to reattach and start moving again (Fig. 3 D). We saw the first fluorescent dynein come through within minutes (3.49 ± 0.12 min, Fig. 3 E). This suggests that at least some dynein is capable of binding cargos and moving them in a highly processive manner along the whole length of the axon.

## LIS1 and NDEL1 undergo long-range retrograde movements along the axon

We next asked if dynein-associated components also undergo long-range movement. Dynactin is required for dynein's processivity (Reck-Peterson et al., 2018) and would therefore be expected to travel long distances. The expectations are less clear for LIS1 and NDEL1, which are involved in the initial formation of motile dynein complexes (Markus et al., 2020). In vitro studies with LIS1 have come to opposing conclusions about whether it can comigrate with dynein complexes (Baumbach et al., 2017; Elshenawy et al., 2020; Gutierrez et al., 2017; Htet et al., 2020; Qiu et al., 2019). To directly visualize retrograde movement of

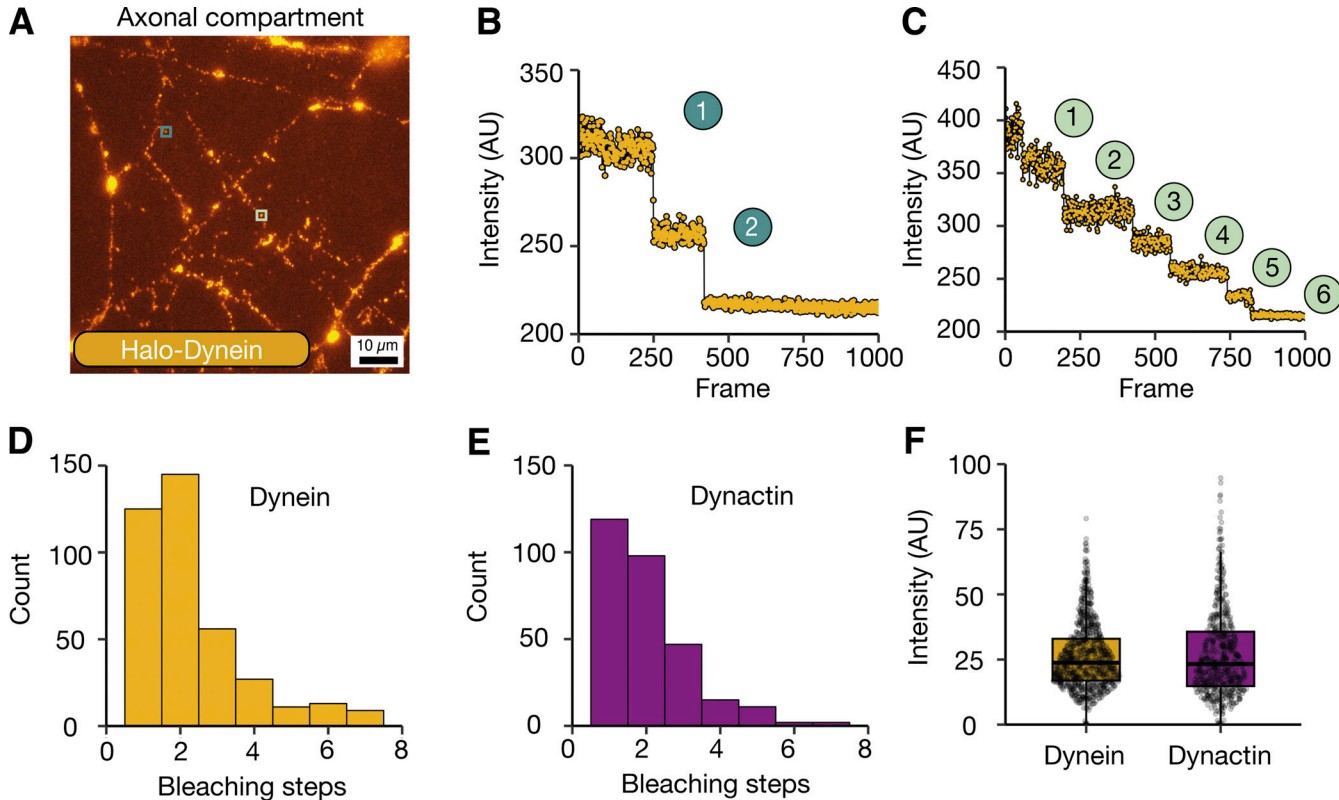

Figure 2. **Visualizing dynein in iNeurons. (A)** Image of 21–23 DPI Halo-*DYNC1H1* iNeuron axons stained with 1 nM JFX 554 and treated with 0.5 µM NEM. Teal and light green insets display spots before bleaching. **(B)** The bleaching trace from teal inset in A. The spot displays two bleaching steps representing the presence of one dynein molecule. **(C)** The bleaching trace from light green inset in A. The spot displays six bleaching steps representing the presence of three dynein molecules. **(D)** The number of bleaching steps from dynein (Halo-*DYNC1H1*) spots (389 spots, 14 videos, N = 3). **(E)** The number of bleaching steps from dynactin (Halo-*ACTR10*) spots (297 spots, 10 videos, N = 3). **(F)** Graph shows the analysis of step size intensity from dynein and dynactin spots during bleaching (dynein: 912 steps, 14 videos, N = 3; dynactin: 623 steps, 10 videos, N = 3). Boxplot shows median, first, and third quartiles. Upper/lower whiskers extend to 1.5× the interquartile range.

these proteins, we used our dynactin cell line (Halo-*ACTR10*, hiPSC) and generated cells with tagged LIS1 (*PAFAB1H1*-Halo, hESCs, heterozygous) and NDEL1 (Halo-*NDEL1*, hESCs, homozygous).

We saw highly processive retrograde events not only with dynactin but also with LIS1 and NDEL1 (Fig. 4 A; and Videos 4, 5, and 6). Although it appeared that there were fewer processive events for both LIS1 and NDEL1 than with dynein and dynactin, the difference was only statistically significant for NDEL1 (LIS1: 1.16 ± 0.12 min[-1], NDEL1: 0.57 ± 0.11 min[-1] versus dynein: 3.00 ± 0.27 min[-1] and dynactin: 3.29 ± 0.66 min[-1], Fig. 4 B). To understand if these proteins had also traveled the length of the axon, we measured how long it took to visualize the first retrograde fluorescent particle to travel through the microfluidic grooves. We found that dynactin, LIS1, and NDEL1 all traveled through the groove in a similar time frame to dynein (Fig. S2 A). This suggests that these proteins bind to a cargo stably throughout their transport along an axon.

Previously, *in vitro* studies showed that when LIS1 is present on dynein complexes, it reduces their speed compared with those without LIS1 (Htet et al., 2020). If this is the case in the axon, we would expect an LIS1 spot to have a lower speed compared with dynein. However, we saw no significant difference in average speed, instantaneous velocity, or pausing

kinetics between any of the dynein machinery (Fig. 4 D and Fig. S2, B–D). This suggests that either LIS1 has a different effect on the dynein motor in the axon or LIS1 travels on cargos without directly interacting with the motors driving transport.

### Dynein and dynactin reach the distal tip of the axon at different speeds

Many organelles are known to move bidirectionally (Maday et al., 2014) and copurify with both kinesin and dynein (Canty et al., 2023; Encalada et al., 2011; Fenton et al., 2021; Maday et al., 2012). Therefore, we expected that both dynein and dynactin would be present on kinesin-driven anterograde vesicles. To test this, we treated dynein, dynactin, LIS1, and NDEL1 iNeurons with 200 nM JFX 554/650 in the somatodendritic compartment and imaged them in the axonal compartment. We observed anterograde movements for all components of the dynein machinery analyzed (Fig. 5 A; and Videos 7, 8, and 9), although there were significantly fewer LIS1 particles compared with dynein and dynactin (LIS1: 1.05 ± 0.10 min[-1], dynein: 2.43 ± 0.15 min[-1], dynactin: 2.87 ± 0.22 min[-1], and NDEL1: 1.76 ± 0.21 min[-1], Fig. 5 B).

Strikingly, we found that the majority of dynein and LIS1 particles traveled at speeds that were significantly faster than

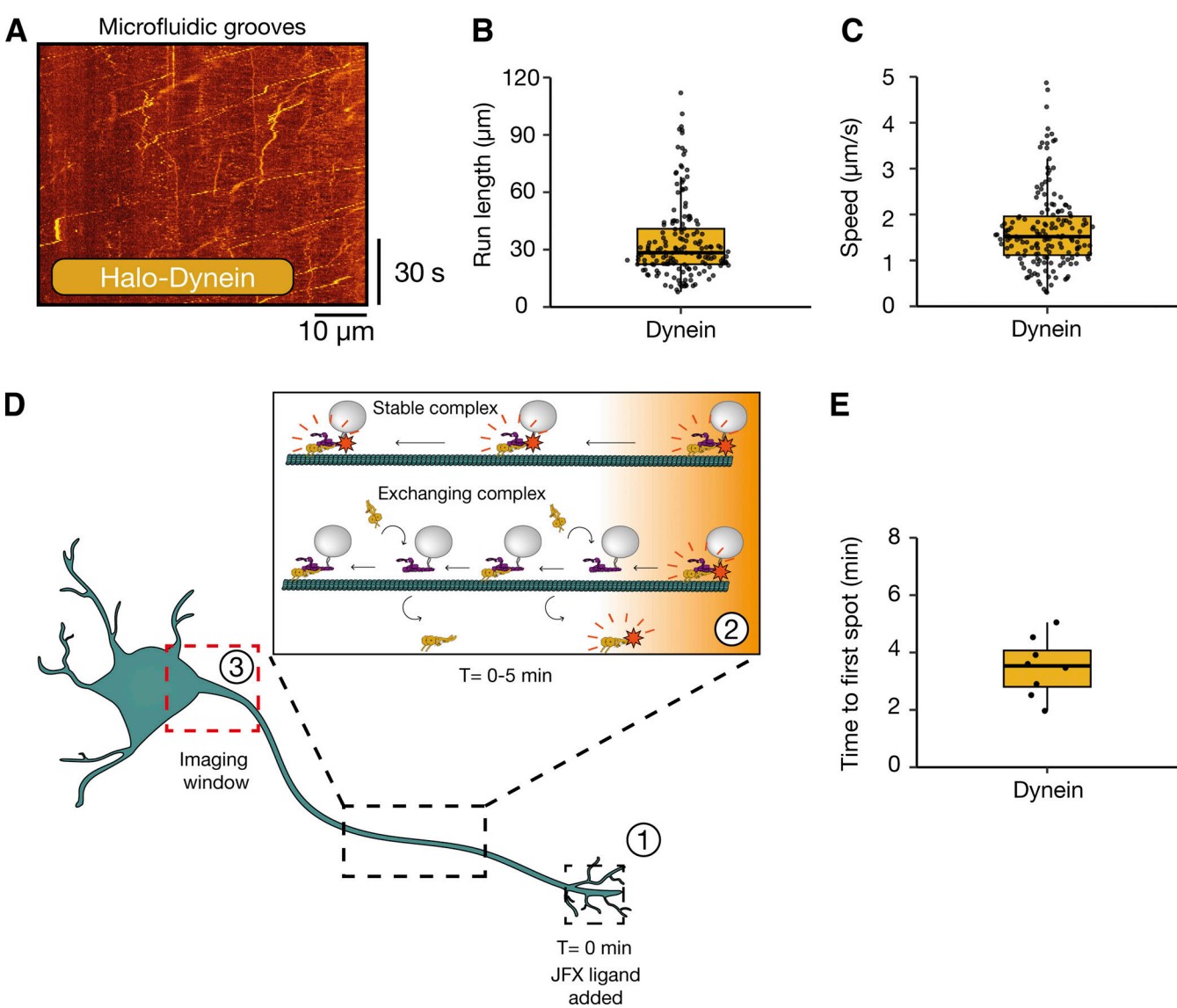

Figure 3. **Dynein moves long-range along the axon in a stable complex. (A)** Example kymograph of retrograde dynein (Halo-*DYNC1H1*) movement in 21–23 DPI neurons treated with either JFX 554 or JFX 650, see also Fig. S3. **(B)** The run lengths of retrograde dynein particles in 21–23 DPI neurons. **(C)** The speed of retrograde dynein particles in 21–23 DPI neurons (162 tracks, 21 videos, N = 6). **(D)** Schematic of the experimental setup to explore dynein movement in the axon. (1) JFX dyes were added to the axon tip at T = 0 min. This labels both diffusive and motile dynein. (2) Motile dynein moves along the axon. Dynein either forms a stable interaction with cargo and moves the entire length of the axon or undergoes short-range movements and then dissociates from the complex. (3) Our imaging window at the proximal end of the axon. If dynein is stably bound, we expected to see the first fluorescent spot within 5 min. On the other hand, if dynein is exchanged on cargo, it should take longer. **(E)** The amount of time until the first processive fluorescent particle was detected in dynein (Halo-*DYNC1H1*) 21–23 DPI neurons (8 videos, N = 8). Boxplots shows median, first, and third quartiles. Upper/lower whiskers extend to 1.5× the interquartile range.

dynactin and NDEL1 (dynein: 3.47 ± 0.04 µm/s and LIS1: 3.67 ± 0.07 µm/s versus dynactin: 1.78 ± 0.03 µm/s and NDEL1: 1.47 ± 0.01 µm/s, Fig. 5 C and Fig. S2 G). LIS1 and dynein also had significantly fewer pauses during their anterograde transport than dynactin and NDEL1 (dynein: 5.25 ± 0.36% and LIS1: 3.30 ± 0.28% versus dynactin: 15.04 ± 0.33% and NDEL1: 20.13 ± 2.04%, Fig. S2, E and F). This suggests dynein and LIS1 are being transported to the axon tip via a different mechanism to dynactin and NDEL1. To explore further, we generated a dual labeled line with both dynein (Halo-*DYNC1H1*) and dynactin (*DCTN4*-SNAP) tagged in the same hESC line. We treated these iNeurons with 200 nM JFX 554 and 1 µM SNAP-SiR to label

dynein or dynactin respectively in the somatodendritic compartment. Again, we found that the majority of dynein moved significantly faster than dynactin in the anterograde direction. However, we saw a small number of colocalized dynein and dynactin particles (~8–12%, Fig. 5, D and E; Fig. S2, H and I; and Video 9). These traveled slightly slower than dynein spots alone but were faster than dynactin spots (dynein: 3.53 ± 0.79 µm/s, dynactin: 1.80 ± 0.49 µm/s, Co-loc: 3.08 ± 0.74 µm/s, Fig. S2 H). Overall, our data suggests that whereas some dynein and dynactin molecules move together retrogradely, the majority are trafficked separately to the distal tip.

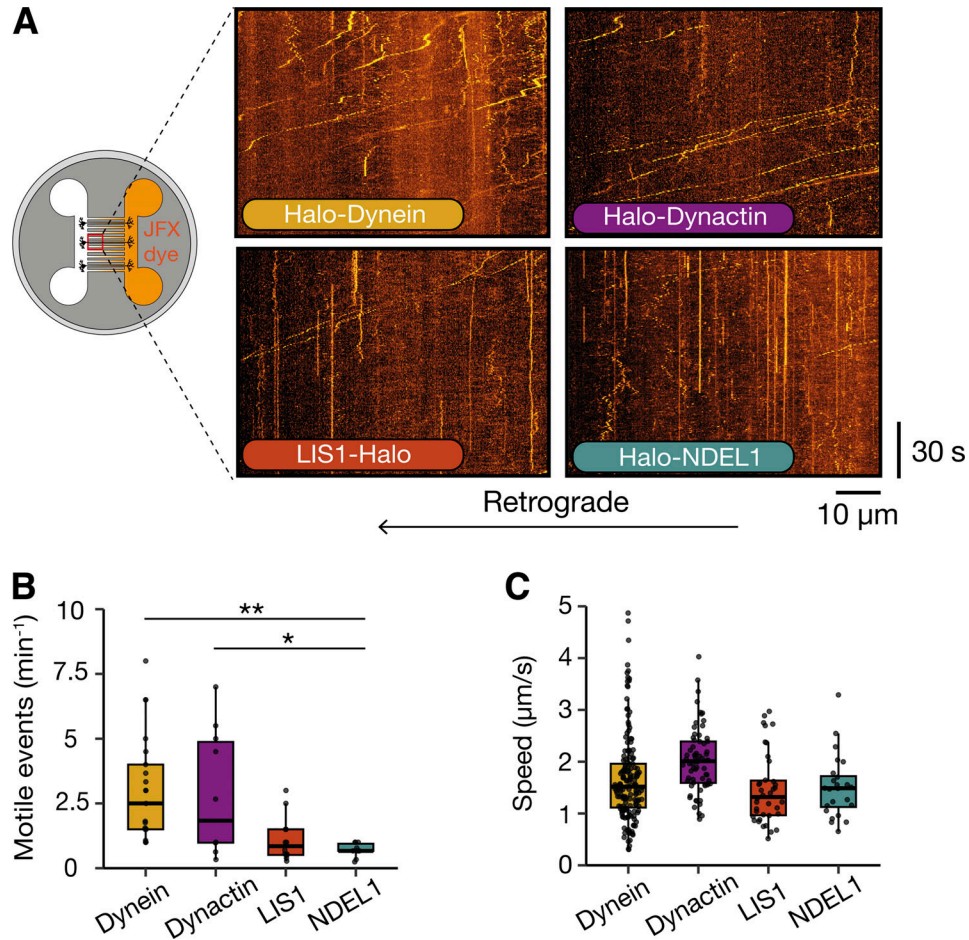

Figure 4. **Dynein machinery moves long-range retrogradely along the axon. (A)** Schematic of microfluidic device showing treatment in the axonal compartment with JFX halo ligand and imaging in the somatodendritic compartment. Example kymographs of retrograde dynein (Halo-*DYNC1H1*), dynactin (Halo-*ACTR10*), LIS1 (*PAFAH1B1*-Halo), and NDEL1 (Halo-*NDEL1*) movement in 21–23 DPI neurons. **(B)** The frequency of retrograde motile events in dynein, dynactin, LIS1, and NDEL1 in 21–23 DPI neurons (dynein: 162 tracks, 21 videos, N = 6; dynactin: 68 tracks, 10 videos, N = 4; LIS1: 38 tracks, 14 videos, N = 6; NDEL1: 24 tracks, 10 videos, N = 3). Dynein versus NDEL1: **P = 0.0088, dynactin versus NDEL1: *P = 0.026, dynein versus dynactin: P = 0.81, LIS1 versus dynactin: P = 0.15, LIS1 versus dynein: P = 0.06, LIS1 versus NDEL1: P = 0.28, Kruskal–Wallis test, Dunn post hoc test. **(C)** Graph showing the speed of retrograde dynein, dynactin, LIS1, and NDEL1 particles in 21–23 DPI neurons. Dynein versus dynactin: P = 0.22, NDEL1 versus dynactin: P = 0.25, LIS1 versus dynactin: P = 0.07, dynein versus NDEL1: P = 0.90, dynein versus LIS1: P = 0.50, NDEL1 versus LIS1: P = 0.68. Boxplots shows median, first, and third quartiles. Upper/lower whiskers extend to 1.5× the interquartile range.

## Dynein movement in the axon

Our data suggested that dynein motors are capable of moving the entire length of the axon. In contrast, a recent study suggested that dynein moves cargo by multiple short runs, detaching after each movement (Tirumala et al., 2024). One explanation is that both mechanisms can be used: with fast moving cargo, such as those visualized in our assays, binding dynein more stably, and slower cargos exchanging dyneins as they travel. Alternatively, transport in HeLa could differ from neurons. To this end, it is notable that in HeLa cells, membrane vesicles undergo lots of short movements interspersed with pauses compared with more continuous long-range transport in neurons (Fellows et al., 2020; Tirumala et al., 2024).

Determining how many dyneins are present on a cargo is challenging in cells. Previous work suggested teams of up to 10 dyneins are required (Rai et al., 2013, 2016). Here, we estimated moving dynein spots that traverse the axon contain 1–4

fluorophores. If fully labeled with the HaloTag, this would correspond to one to two dynein molecules, agreeing with previous data on dynein numbers on endosomes (Tirumala et al., 2024). Our data on NEM-treated static spots showed a wider distribution of dynein numbers with up to eight fluorophores present (Fig. 2 B). An explanation for the difference in numbers might be the size of the cargo. The nature of our experiments selects for the fastest cargos in the axon, and we know from previous work that smaller cargos, such as endosomes, which have room for fewer motors, move faster (Fellows et al., 2020) (Fig. 1 D). In contrast, larger cargos such as lysosomes are suggested to require approximately eight dyneins (Ori-Mckenney et al., 2010). Another explanation could be that some dynein dissociates from our cargos during transit. Despite this, our results suggest at least some dyneins remain on cargos throughout transport along the axon.

Some vesicles are known to mature and change their composition during their transport in the axon (Cason and Holzbaur,

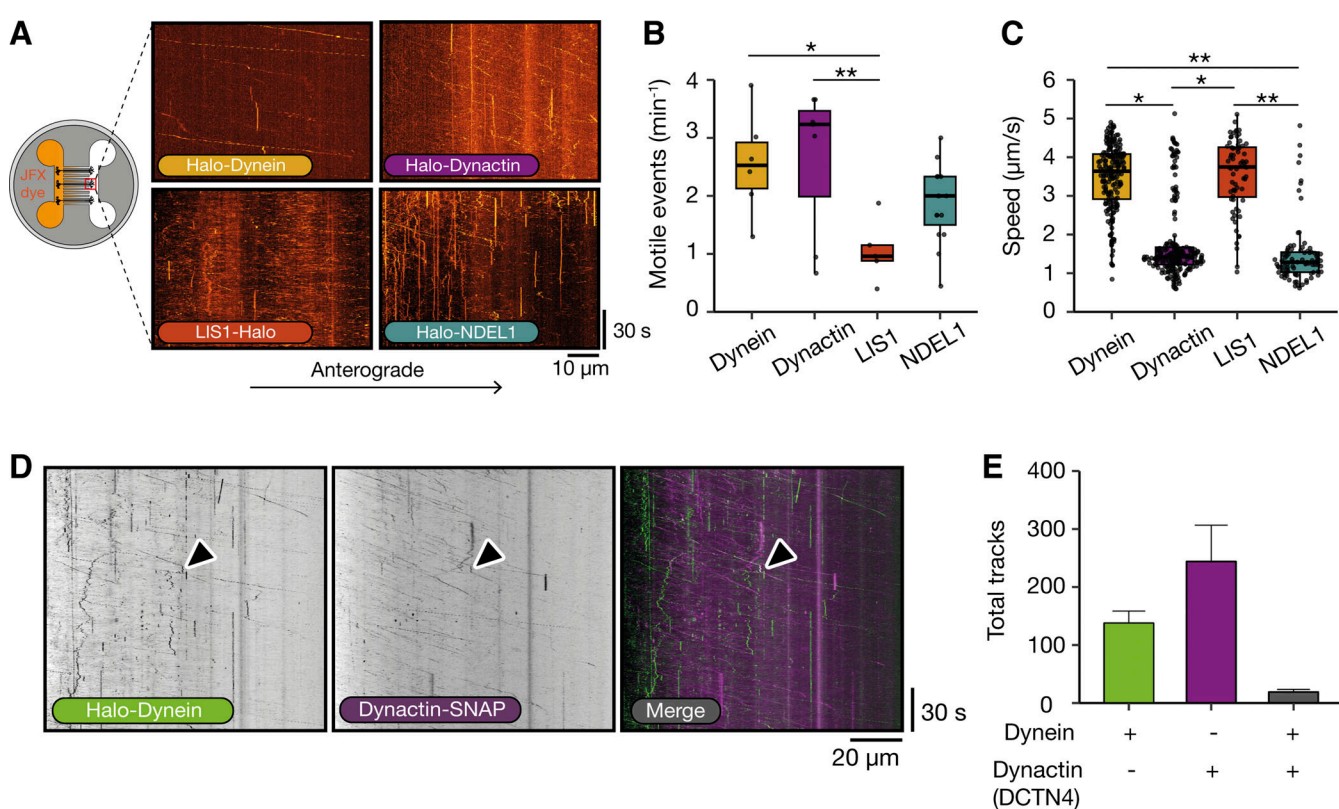

Figure 5. **Dynein and its machinery travel to the distal tip of the axon separately. (A)** Schematic of microfluidic device showing treatment in the somatodendritic compartment with JFX 554/650 ligand and imaging in the axonal compartment. Example kymographs of retrograde dynein (Halo-*DYNC1H1*), dynactin (Halo-*ACTR10*), LIS1 (*PAFAH1B1*-Halo), and NDEL1 (Halo-*NDEL1*) movement in 21–23 DPI neurons. **(B)** The frequency of anterograde motile events in dynein, dynactin, LIS1, and NDEL1 in 21–23 DPI neurons (dynein: 198 tracks, 6 videos, N = 5; dynactin: 211 tracks, 7 videos, N = 5; LIS1: 68 tracks, 5 videos, N = 5; NDEL1: 86 tracks, 15 videos, N = 3). Dynein versus LIS1: *P = 0.033, dynactin versus LIS1: **P = 0.0077, dynein versus dynactin: P = 0.60, dynactin versus NDEL1: P = 0.17, dynein versus NDEL1: P = 0.36, LIS1 versus NDEL1: P = 0.36, Kruskal–Wallis test, Dunn post hoc test. **(C)** The speed of anterograde dynein, dynactin, LIS1 and NDEL1 particles in 21–23 DPI neurons (dynein: 198 tracks, 6 videos, N = 5; dynactin: 211 tracks, 7 videos, N = 5; LIS1: 68 tracks, 5 videos, N = 5; NDEL1: 86 tracks, 15 videos, N = 3). Dynein versus dynactin: *P = 0.044; dynein versus NDEL1: **P = 0.0056; dynactin versus LIS1: *P = 0.015; LIS1 versus NDEL1: **P = 0.0018, dynein versus LIS1: P = 0.68, dynactin versus NDEL1: P = 0.30, Kruskal–Wallis test, Dunn post hoc test. **(D)** Example kymographs of anterograde dynein (Halo-*DYNC1H1*) and dynactin (*DCTN4*-SNAP) movement in 21–23 DPI neurons. Arrows point to colocalized dynein and dynactin. **(E)** The average number of tracks per experiment in 21–23 DPI neurons (dynein: 413 tracks, 25 videos, N = 3; dynactin: 732 tracks, 25 videos, N = 3; Co-loc: 57 tracks, 25 videos, N = 3). Boxplot shows median, first, and third quartiles. Upper/lower whiskers extend to 1.5× the interquartile range.

2022; Kulkarni and Maday, 2018). Recent work on autophagosomes suggested that different cargo-specific adaptors are required for dynein transport in different segments of the axon (Cason et al., 2021). An interesting question is how these observations relate to the very long-distance movement of dynein. One possibility is that additional dynein molecules are tethered onto cargos independently of the activating adaptor. In this way, the pool of dynein that moves along the axon would engage different adaptors when required. Another possibility is that some cargos engage dynein for the whole duration of their transport whereas others, which we would not detect in the experiments reported here, show exchange of both motors and adaptors.

### LIS1 and NDEL1 move retrogradely

LIS1 and NDEL1 are integral for dynein transport (Lam et al., 2010). LIS1 helps initiate dynein movement (Elshenawy et al., 2020; Htet et al., 2020; Qiu et al., 2019) by disrupting its autoinhibition and supporting the formation of active complexes with dynactin (Elshenawy et al., 2020; Gillies et al., 2022; Htet

et al., 2020; Marzo et al., 2020; Qiu et al., 2019; Singh et al., 2023, *Preprint*). On the other hand, NDEL1 recruits LIS1 to dynein (Garrott et al., 2023, *Preprint*; Okada et al., 2023, *Preprint*). Whether these proteins remain part of the motile complex was unclear. Some data suggest that LIS1 co-migrates with dynein/dynactin complexes *in vitro* (Baumbach et al., 2017; Gutierrez et al., 2017; Jha et al., 2017), whereas others studies find LIS1 dissociates from moving complexes (Egan et al., 2012; Elshenawy et al., 2020; Gillies et al., 2022; Htet et al., 2020). However, this issue had not been addressed in mammalian cells. Our finding that LIS1 and NDEL1 are both transported long distances raised the question of why they comigrate with cargos. One possibility is that vesicles contain both actively engaged dynein/dynactin and reserve motors. In this case, the presence of LIS1 would allow the formation of new active dynein/dynactin complexes during a cargo's journey along the axon. New initiation events may be required when cargos encounter obstacles. This was highlighted by a study where LIS1 and NDEL1 were shown to facilitate increased force production of dynein when

cargo movement was restrained by an optical trap (Reddy et al., 2016). Overall, we find both proteins are transported along the length of the axon and may therefore play a role in long-range trafficking.

### Dynein and its machinery move to the axon tip at different speeds

Our observation that dynein and LIS1 travel much faster (~3.5 µm/s) in the anterograde direction than dynactin and NDEL1 (~1.6 µm/s) raises the question of how these proteins are transported at different speeds. Dynein, dynactin, and LIS1 have all been found to bind the plus end of microtubules (Carvalho et al., 2004; Vaughan et al., 1999); however this speed is ~0.1 µm/s (Fellows et al., 2020) and therefore cannot account for our observed movement. Instead, it is likely the two groups are both driven by kinesin motors. Kinesin-1, -2, and -3 are the major anterograde motors present in neurons (Sleigh et al., 2019). Certain kinesins are faster than others, with kinesin-3 being roughly three times faster than kinesin-1 (Lipka et al., 2016). Therefore, dynein/LIS1 and dynactin/NDEL1 may be driven by different kinesins. Another mechanism would be to alter the speed of the same kinesin when it is bound to different cargos. For example, neuronal APP (amyloid precursor protein) vesicles, which are driven by kinesin-1, move much faster (~3.6 µm/s) than other kinesin-1 cargos (Araki et al., 2007). This faster speed depends on the presence of the adaptor protein JIP1 (Tsukamoto et al., 2018) and matches that of our anterograde moving dynein and LIS1.

### Implications of transporting dynein and dynactin separately to the axon tip

Several studies suggest that kinesin and dynein are both present on cargos and can alternate activity to cause rapid reversals in direction (Hancock, 2014). Lines of evidence include colocalization of both motors on cargo (Encalada et al., 2011; Maday et al., 2012), the ability of both motors to simultaneously bind to the same adaptor proteins (Canty et al., 2023; Fenton et al., 2021; Kendrick et al., 2019), and observations that inhibition of either motor leads to bidirectional transport defects (Ally et al., 2009; Encalada et al., 2011; Martin et al., 1999; Sainath and Gallo, 2015). As dynactin is known to be required for dynein function, we had assumed that it would travel with dynein in both the retrograde and anterograde directions. Although we saw some dynactin moving anterogradely at the same speed as dynein, the majority moved slower. What could explain this predominant separation of dynein and dynactin?

One possibility is that the missing component, either dynein or dynactin, is picked up in transit resulting in a reversal. Similar examples include in *Ustilago maydis,* where kinesin cargos reverse upon meeting a dynein moving in the other direction. In this case, however, the reversal is likely due to the recruitment of both dynein and dynactin (Bielska et al., 2014). More recently, work in HeLa cells suggested that dynactin, adaptors, and cargos wait on microtubules and only move when dynein is recruited (Tirumala et al., 2024), although the situation may be different in neurons where cargos are moving much longer distances.

An alternative explanation is that neurons separate dynein and dynactin for rapid delivery to the axon tip. Previous work showed that ~90% of dynein molecules are moved anterogradely by slow axonal transport (Dillman et al., 1996a, 1996b). This process is driven by transient, direct interactions between dynein and kinesin-1 (Twelvetrees et al., 2016). What role do these fast and slow pools of dynein then have in the neuron? It has been suggested that the rapid delivery of specific dynein isoforms may be needed for retrograde transport in the neuron whilst the slow pool is important for additional functions (Susalka et al., 2000). Our observations of a large flux of fast anterograde-moving dynein and dynactin movement are consistent with this hypothesis. The separate movement of dynein and dynactin would have the advantage that they are less likely to be activated inappropriately. Likewise, the separate anterograde movement of the initiation factors NDEL1 and LIS1 would also ensure that the retrograde transport machinery predominantly assembles where it is needed at the axon tip. This agrees with a previous study that suggested that LIS1 holds dynein in an inhibited form allowing it to be moved anterogradely and that NDEL1 is responsible for releasing the inhibition at the distal tip (Yamada et al., 2008). Taken together, our study highlights a highly efficient transport system with microtubule motors only active when needed. This could be particularly important in axons where many cargos move predominantly long-range.

## Materials and methods

### Human stem cell culture and NGN2 neuronal differentiation

hESC (H9 line; WiCELL) and hiPSC (Bit Bio Ltd) (Pawlowski et al., 2017), which harbor a doxycycline-inducible NGN2 transgene in the AAVS1 locus, were kept on Cultrex basement membrane extract (35 µg/cm², 3432-010-01; R&D systems) and fed every other day with mTeSR plus (100-0276; STEMCELL Technologies). Cells were kept at 37°C in a 5% $CO_2$ incubator.

To differentiate into iNeurons, cells were dissociated into single cells with accutase (07920; STEMCELL Technologies), and 300,000 cells were plated per well of a Cultrex-coated six-well dish. For the first 24 h, cells were kept in mTeSR plus and CloneR2 (1×; 100-0691, STEMCELL Technologies). After that, media was switched to differentiation media (DMEM/F12, (11330032; Gibco), GlutaMAX (1×, 35050038; Gibco), Non-Essential Amino Acids (1×, 11140-50; Gibco), N2 supplement (1×, 175020-48; Gibco), Penicillin/Streptomycin (1%), and doxycycline (1 µg/ml, D9891; Merck)) for 48 h. After this time, cells were dissociated with accutase and immediately plated into microfluidics (PDMS mould on glass bottom dish (HBST-5040, #1.5H, 0.005 mm; Willco Wells)) containing neuronal media (Neurobasal, 21103049; Gibco), GlutaMAX (1×), B27 supplement (17504044; Gibco), BDNF (450-02-50UG, 10 ng/ml; Peprotech), NT3 (450-03-50UG, 10 ng/ml; Peprotech), Penicillin/Streptomycin (1%), and doxycycline (1 µg/ml)). Microfluidics were coated with poly-D-lysine (A38904-01, 20 µg/ml; Merck) and Geltrex hESC-Qualified reduced growth factor basement membrane matrix (A15696-01, 0.12–0.18 mg/ml; Thermo Fisher Scientific). A 25% media exchange took place every 2 d until 7 DPI where doxycycline was removed from the neuronal media. iNeurons were cultured until 21–23 DPI when imaging took place. Cells were kept at 37°C in a 5% $CO_2$ incubator.

## CRISPR knock-in of the HaloTag to stem cells

Knock-in of the HaloTag to hESC or hiPSCs was done following established methods (Bruntraeger et al., 2019). Briefly, ribonucleoprotein (RNP) complexes were formed with 1 µl HiFi Cas9 (1081061, 4 µg/µl; IDT), 6 µl synthetic sgRNA (30 µM, with the following protospacer sequences *DYNC1H1*: 5′-CTCCGACATGGT GTCGCGCT-3′, *ACTR10*: 5′-CGTAGAGCGGCATGGTAGTA-3′, *PAFAB1H1*: 5′-GCCGTTGATTGTGTCTCCTT-3′, *NDEL1*: 5′-TTC ACAGGCTTTCTTGATCA-3′, *DCTN4*: 5′-CCCTCCAGTGGAACC TT-3′, Synthego) and nucleofection buffer P3 (V4XP-3032; Lonza). ssDNA (6 µg) containing 100–150 nt homology arms flanking the HaloTag or SNAP-tag coding sequence were added. 215,000 accutase-dissociated hESC or hiPSC were mixed with RNP complexes and nucleofected with the 4D-Nucleofector (program CA-137; Lonza). The cells were then plated in a six-well dish coated with rhLaminin-521 (0.5 µg/cm²; Gibco) with mTeSR plus and CloneR. Cells were left until confluent (~7 d). At this time point, the cells were treated with 200 nM JF646 Halo-Tag ligand (GA1120; Promega) for 20 min. Cells were then accutase-dissociated and washed twice in 4 ml PBS. Halo-positive cells were then flow-sorted and 3,000 Halo+ cells were plated at clonal density on a Cultrex-coated 10-cm dish. Cells were kept in mTeSR plus and CloneR. Individual colonies were picked and screened for successful gene editing by PCR and Sanger sequencing (Fig. S3). The ACTR10 (iPSC) homozygous line was made at the Wellcome Sanger Institute by Dr. Andrew Bassett. Knock-in lines were then differentiated into iNeurons following the above protocol.

## Immunofluorescence

iNeurons at 21–23 DPI were fixed in 4% paraformaldehyde (28908; Sigma-Aldrich) in PBS for 12 min at room temperature. Cells were then washed with PBS, permeabilized, and blocked for 15 min in a permeabilization buffer (0.5% bovine serum albumin (9048-46-8); 10% donkey serum (C06SB); 0.2% Triton X-100 (9036-19-5); in PBS). Primary antibodies against SMI-31P (801601, 1:500; BioLegend), MAP2 (188004, 1:500; synaptic systems), and β3-tubulin (T2200, 1:1,000; Sigma-Aldrich) were diluted in blocking buffer (0.5% BSA, 10% donkey serum in PBS) and incubated with cells for 1 h at room temperature. iNeurons were then washed three times with PBS and incubated with fluorescently conjugated secondary antibodies (Goat anti-Guinea pig IgG [H+L] Alexa Fluor 488 (A11073, 1:1,000; Invitrogen), donkey anti-Rabbit IgG [H+L] Alexa Fluor 555 (A31572, 1:1,000; Invitrogen), and donkey anti-Mouse IgG [H+L] Alexa Fluor 647 (A31571, 1:1,000; Invitrogen)) diluted in blocking buffer for 1 h. Cells were washed with PBS and mounted with ProLong Diamond antifade mountant (P36961; Thermo Fisher Scientific). Finally, iNeurons were imaged using an inverted Zeiss LSM 780 using a 63×, 1.4 NA DIC Plan-Apochromat oil immersion objective.

## Photobleaching step analysis

Halo-*DYNC1H1* and Halo-*ACTR10* iNeurons were cultured in microfluidics until 21–23 DPI. Cells were treated with 1 nM JFX Halo ligand in the axonal chamber for 20 min. The ligand was washed out with new neuronal medium and cells were left overnight at 37°C in a 5% $CO_2$ incubator. Cells where then treated with 0.5 µM N-ethyl maleimide (23030, NEM; Thermo Fisher Scientific) for 20 min. Cells were imaged at 37°C using an inverted Nikon 100×, 1.49 NA CFI Apochromat oil immersion TIRF lens. To improve the signal-to-noise ratio, cells were imaged using HILO imaging (Tirumala et al., 2024). HILO settings were optimized for each condition by altering the angle of incidence of the excitation laser (561 or 640 nm) between 57°–60°. Laser power was kept constant at 50% (15 mW at fiber, LU-N4; Nikon). Time-lapse images were acquired at 30 Hz with 30 ms exposure (sCMOS, 95% QE, Prime 95b; Teledyne Photometrics) continuously for 2 min. Cells were kept at 37°C with 5% $CO_2$ (stage-top incubator, OkoLabs). Spots were picked using ImageJ, and intensity analysis was run using custom scripts in Matlab at https://github.com/carterlablmb.

## Live-cell imaging

Live imaging of iNeurons took place between 21 and 23 DPI. Endosome, lysosome, and mitochondrial transport were assessed in microfluidics with the addition of either 1 µg/ml CTB AlexaFluor 488 (C34775; Thermo Fisher Scientific), 50 nM Lysotracker Deep Red (L12492; Thermo Fisher Scientific), or 100 nM Mitotracker Deep Red FM (M22426; Thermo Fisher Scientific) to the axonal compartment for 30 min at 37°C. Cells were washed and then new prewarmed low fluorescent BrainPhys (05796; STEMCELL Technologies) supplemented with Gluta-MAX (1×; Gibco), B27 supplement (1×; Gibco), BDNF (10 ng/ml; Peprotech), NT3 (10 ng/ml; Peprotech), Penicillin/Streptomycin (1%) was added to cells. 15 min later, the transport was imaged at 37°C using an inverted Zeiss LSM 880 using a 63×, 1.4 NA DIC Plan-Apochromat oil-immersion objective. Images were taken at 2 Hz over a period of 2–4 min.

For endogenous HaloTag/SNAP-tag imaging, iNeuron media was exchanged to prewarmed low fluorescent BrainPhys medium (STEMCELL Technologies) supplemented with GlutaMAX (1×; Gibco), B27 supplement (1×; Gibco), BDNF (10 ng/ml; Peprotech), NT3 (10 ng/ml; Peprotech), Penicillin/Streptomycin (1%). We first treated cells with increasing concentrations of Halo ligand (1–500 nM) to assess which led to the best labeling. We determined that 200 nM was sufficient. Therefore, cells were treated with either 200 nM JFX 554/650 (Grimm et al., 2021) in the axonal (Retrograde: Fig. 3 and Fig. S2) or 200 nM JFX 554/650 or 1 µM SNAP-SiR in the somatodendritic compartment (Anterograde: Fig. 5 and Fig. S2). Cells were imaged immediately at either the somatodendritic compartment (Retrograde: Fig. 3 and Fig. S2) or axonal compartment (Anterograde: Fig. 5 and Fig. S2) of the microfluidic at 37°C (stage-top incubator, OkoLabs). This was done using an inverted Nikon 100×, 1.49 NA CFI Apochromat oil immersion TIRF lens. To improve the signal-to-noise ratio, cells were imaged using HILO imaging (Tirumala et al., 2024). HILO settings were optimized for each condition by altering the angle of incidence of the excitation laser (561 or 640 nm) between 57° and 60°. Laser power was kept constant at 50% (15 mW at fiber, LU-N4; Nikon). Time-lapse images were acquired at 2 Hz with 30 ms exposure (sCMOS, 95% QE, Prime 95b; Teledyne Photometrics) for 2–15 min.

## Dual imaging of HaloTag and SNAP-tag

For dual imaging of the Halo-*DYNC1H1/DCTN4*-SNAP iNeuron line, iNeuron media was exchanged to prewarmed low fluorescent BrainPhys medium (STEMCELL Technologies) supplemented with GlutaMAX (1×; Gibco), B27 supplement (1×; Gibco), BDNF (10 ng/ml; Peprotech), NT3 (10 ng/ml; Peprotech), and Penicillin/Streptomycin (1%). Cells were first treated with 1 μM SNAP-SiR in the somatodendritic compartment for 20 min. 200 nM JFX 554 was then added to the somatodendritic compartment. 5 min later, cells were imaged in the axonal compartment of the microfluidic at 37°C (stage-top incubator, Oko Labs). This was done using an inverted Nikon 100×, 1.49 NA CFI Apochromat oil immersion TIRF lens. Laser power was kept constant at 80% (561 nm) or 100% (640 nm) (15 mW at fiber, LU-N4; Nikon) with a Semrock BLP01-635R-25 long-pass filter. Time-lapse images were acquired at 4 Hz with 50 ms exposure (sCMOS, 95% QE, Prime 95b; Teledyne Photometrics) for 3 min. To improve the signal-to-noise ratio, we used the NOISE2VOID (N2V) (Krull, 2019) denoising convolutional neural network (Video 10). Models were trained for both Halo-*DYNC1H1* (JFX 554) and *DCTN4*-SNAP (SiR) and then used on all images.

## Analysis and quantification

Image analysis was done using Fiji (NIH). To quantify organelle and endogenous Halo/SNAP-tagged protein kinetics, Trackmate imaging software was used (Ershov et al., 2022; Tinevez et al., 2017). For this analysis, spots were tracked frame by frame using the manual tracking implementation, and the output data consisted of spot, edge, and track files. The edges files contain data of spot displacement from frame to frame, whereas the track files summarize the overall movement of the spots from start to finish. An example video with tracked spots has been included and compared with a kymograph made from the same file (Video 3 and Fig. S3). In the text, speed refers to the average speed of spots from the beginning to the end of their track. Only spots that moved over 10 μm were analyzed further. Pauses were defined as the particles moving slower than 0.1 μm/s.

## Statistical analysis

All analysis was undertaken in R (R Core Team, 2014). Data were assessed for normality by Shapiro–Wilk test. To assess the difference between the two groups with a normal distribution, a Student's $t$ test was used. For analysis of multiple groups, either a one-way ANOVA followed by Tukey's multiple comparison test was used or if the data was not normally distributed, the Kruskal–Wallis test was used followed by the Dunn test. All statistics were done on the mean value from each biological replicate. Statistical significance is noted as follows: *$P \leq 0.05$, **$P \leq 0.01$, and ***$P \leq 0.001$. All statistical tests and associated P values are indicated in the figure legends.

## Online supplemental material

Fig. S1 shows additional data for Figs. 1, 2, and 3. Fig. S2 shows additional data for Figs. 4 and 5. Fig. S3 shows an example of tracking methodologies and genotyping data for the cell lines. Video 1 shows imaging of Halo-*DYNC1H1* DPI 21–23 iNeurons treated with 1 nM JFX 650 in the axonal compartment. Imaged at 30 fps. Playback 60 fps. Video 2 shows photobleaching in Halo-DYNC1H1 DPI 21–23 iNeurons in the axonal compartment treated with 1 nM JFX 554 and 0.5 μM NEM. Imaged at 30 fps. Playback 60 fps. Video 3 shows imaging of Halo-DYNC1H1 retrograde movement in DPI 21–23 iNeurons treated with 200 nM JFX 554. Imaged at 2 fps. Playback 20 fps. Includes tracking from Trackmate Fiji plugin. Video 4 shows imaging of Halo-ACTR10 retrograde movement in DPI 21–23 iNeurons treated with 200 nM JFX 554. Imaged at 2 fps. Playback 20 fps. Video 5 shows imaging of Halo-PAFAH1B1 retrograde movement in DPI 21–23 iNeurons treated with 200 nM JFX 554. Imaged at 2 fps. Playback 20 fps. Video 6 shows imaging of Halo-NDEL1 retrograde movement in DPI 21–23 iNeurons treated with 200 nM JFX 554. Imaged at 2 fps. Playback 20 fps. Video 7 shows imaging of Halo-PAFAH1B1 anterograde movement in DPI 21–23 iNeurons treated with 200 nM JFX 554. Imaged at 2 fps. Playback 20 fps. Video 8 shows imaging of Halo-NDEL1 anterograde movement in DPI 21–23 iNeurons treated with 200 nM JFX 554. Imaged at 2 fps. Playback 20 fps. Video 9 shows imaging of dual-labeled Halo-DYNC1H1 and DCTN4-SNAP 21–23 DPI iNeurons. Treated with 200 nM JFX 554 and 1 μM SiR-SNAP. Imaged at 4 fps. Images have been denoised with N2V. Playback 20 fps. Video 10 shows Example of Noise2Void denoising on *DCTN4*-SNAP 21–23 DPI iNeurons treated with 1 μM SiR-SNAP. Original video (top) and denoised video (bottom). Imaged at 4 fps. Playback 20 fps.

## Data availability

All data and analysis files are available at https://doi.org/10.5281/zenodo.8082407. The scripts used for photobleaching analysis can be found at https://github.com/carterlablmb.

## Acknowledgments

We thank J. O'Neil and N. Rzechorzek for help setting up the iNeurons; University of Cambridge, Mark Kotter lab for the provision of the neuron inducible human pluripotent stem cell line (NGN2-OPTi-OX); and S. Bullock for helpful discussions, reading, and help with conceiving the project. R. Wademan and G. Manigrasso for help with cell culture; S. Chaaban for critical reading of the manuscript; J. Grimmett and T. Darling for providing scientific computing resources; and finally, we thank the light microscope and flow cytometry core facilities at the MRC Laboratory of Molecular Biology for experimental and technical assistance.

This work was supported by Wellcome (210711/Z/18/Z), the Medical Research Council, as part of UK Research and Innovation (MRC file reference number MC_UP_A025_1011). For the purpose of open access, the author has applied a CC BY public copyright license to any author-accepted manuscript version arising. Open Access funding provided by MRC Laboratory of Molecular Biology.

Author contributions: A.D. Fellows performed the experiments and analyzed and prepared the figures. A.D. Fellows, M. Bruntraeger, T. Burgold, and A.R. Bassett generated the knock-in cell lines used in this manuscript. A.D. Fellows and A.P. Carter conceived the project and wrote the manuscript.

Disclosures: All authors have completed and submitted the ICMJE Form for Disclosure of Potential Conflicts of Interest. A.R.

Bassett reported personal fees from Ensocell outside the submitted work; in addition, A.R. Bassett had a patent to one step RMCE tagging issued. No other disclosures were reported.

Submitted: 20 September 2023

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

# Supplemental material

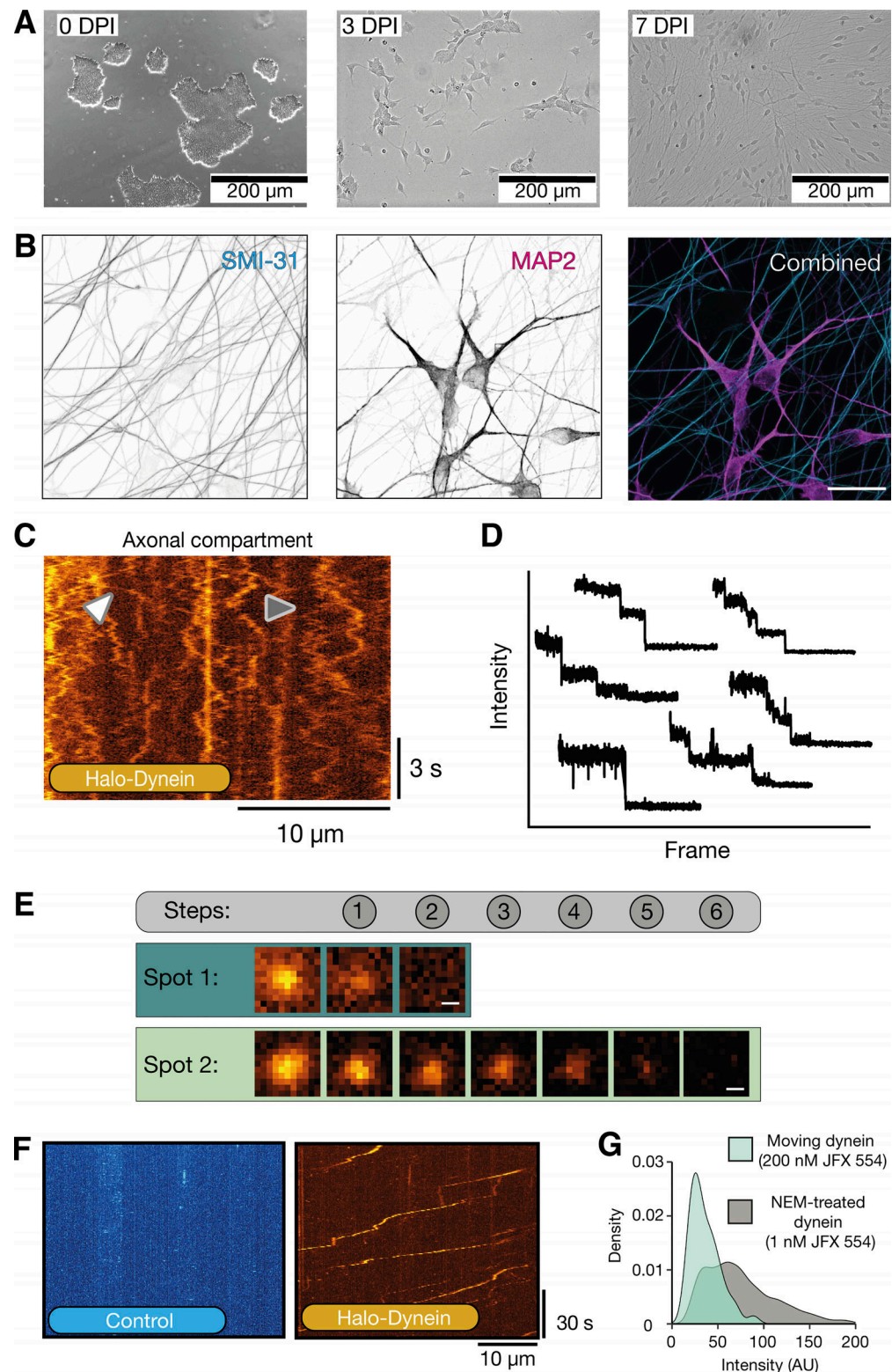

Figure S1.   **iNeurons as a model to study dynein mediated transport. (A)** Example immunofluorescence images of different stages of iNeuron differentiation. DPI 0 = stem cells, DPI 3 = 3 days post induction with doxycycline, DPI 7 = 7 days post induction with doxycycline. **(B)** Image of 21–23 DPI iNeurons showing staining with axonal (SMI-31, cyan) and dendritic (MAP-2, magenta) markers. Scale bar is 20 µm. **(C)** Example kymograph of 21–23 DPI Halo-*DYNC1H1* treated with 1 nM JFX 554. White arrow highlights processive event. Gray arrow shows diffusive motility. **(D)** 6 bleaching traces from dynein spots. The traces were separated to enhance clarity. **(E)** Images of two spots during bleaching from Fig. 2, B and C. The teal spot undergoes 2 bleaching steps and corresponds to the bleaching trace in Fig. 2 B. The light green spot undergoes 6 bleaching steps and corresponds to Fig. 2 C. Scale bar is 0.3 µm. **(F)** Example kymographs of control (untagged) and Halo-*DYNC1H1* 21–23 DPI iNeurons treated with 500 nM JFX 554. **(G)** The intensity of moving dynein spots versus dynein spots treated with NEM (Fig. 2).

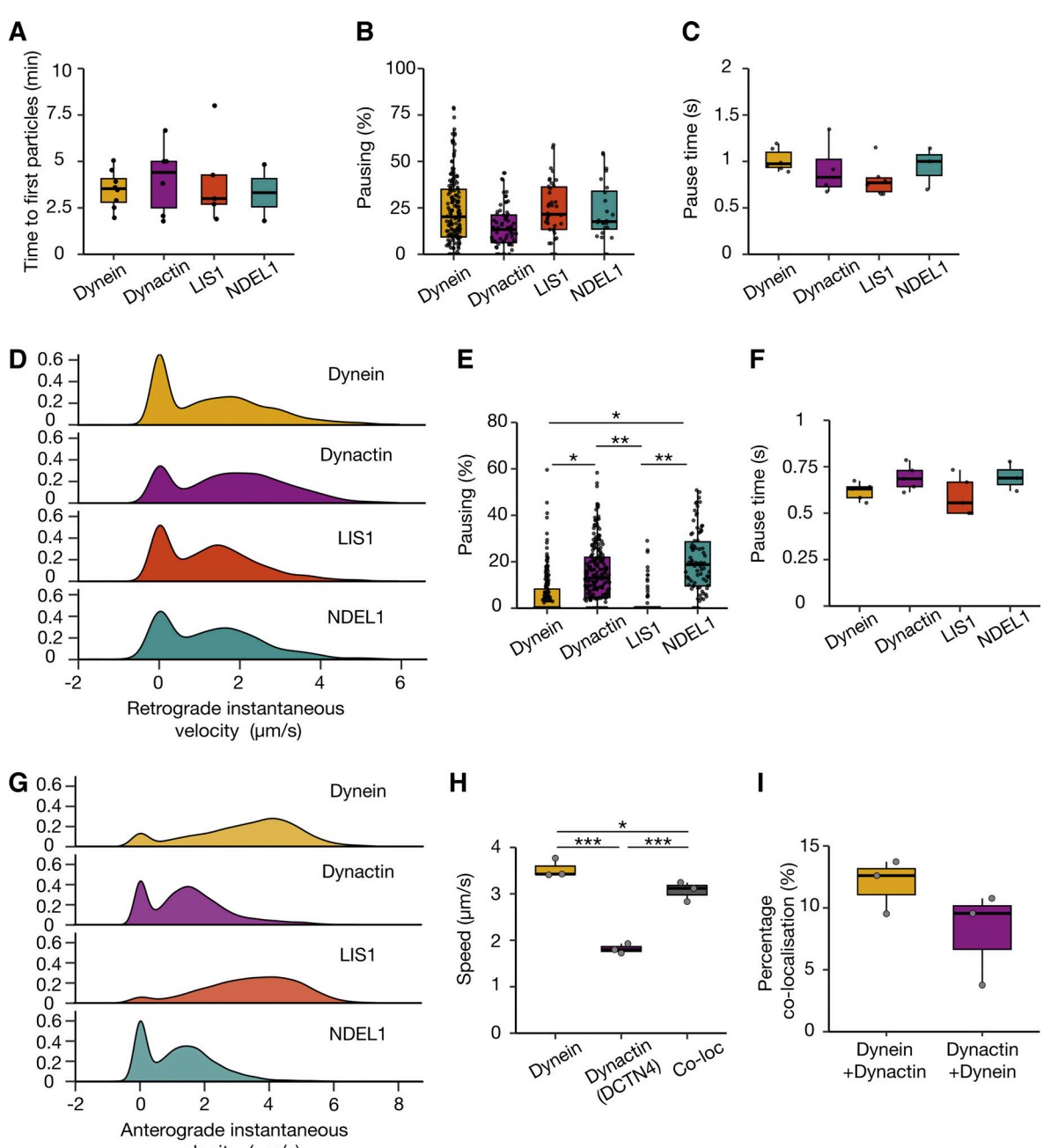

Figure S2. **Analysis of the dynein machinery in the axon. (A)** The amount of time until the first processive retrograde fluorescent particle was detected in dynein, dynactin, LIS1, and NDEL1 21–23 DPI neurons (dynein: 8 videos, $N$ = 8; dynactin: 6 videos, $N$ = 6; LIS1: 5 videos, $N$ = 5; NDEL1: 2 videos, $N$ = 2). Dynein versus dynactin: P = 0.67, NDEL1 versus dynactin: P = 0.53, LIS1 versus dynactin: P = 0.72, dynein versus NDEL1: P = 0.72, dynein versus LIS1: P = 0.99, NDEL1 versus LIS1: P = 0.73. **(B)** The amount of time dynein, dynactin, LIS1, and NDEL1 spent pausing in the retrograde direction (dynein: 162 tracks, 21 videos, $N$ = 6; dynactin: 68 tracks, 10 videos, $N$ = 4; LIS1: 38 tracks, 14 videos, $N$ = 6; NDEL1: 24 tracks, 10 videos, $N$ = 3). Dynein versus dynactin: P = 0.18, NDEL1 versus dynactin: P = 0.42, LIS1 versus dynactin: P = 0.09, dynein versus NDEL1: P = 0.71, dynein versus LIS1: P = 0.68, NDEL1 versus LIS1: P = 0.48. Pauses are defined as spots moving slower than 0.1 µm/s. **(C)** The average length of time each pause lasted for dynein, dynactin, LIS1, and NDEL1. Dynein versus dynactin: P = 0.3, NDEL1 versus dynactin: P = 0.63, LIS1 versus dynactin: P = 0.50, dynein versus NDEL1: P = 0.68, dynein versus LIS1: P = 0.06, NDEL1 versus LIS1: P = 0.26. Kruskal–Wallis test, Dunn post hoc test. **(D)** Instantaneous retrograde velocities of dynein, dynactin, LIS1, and NDEL1 particles in 21–23 DPI neurons. **(E)** The percent of time dynein, dynactin, LIS1, and NDEL1 spent pausing. Dynein versus dynactin: *P = 0.044; dynein versus NDEL1: *P = 0.026; dynactin versus LIS1: **P = 0.0037; LIS1 versus NDEL1: **P = 0.0028, dynactin versus NDEL1: P = 0.63, dynein versus LIS1: 0.37. Kruskal–Wallis test, Dunn post hoc test. Pauses are defined as spots moving slower than 0.1 µm/s. **(F)** Average length of time each pause lasted for dynein, dynactin, LIS1, and NDEL1 in the anterograde direction. Dynein versus dynactin: P = 0.15, NDEL1 versus dynactin: P = 0.95, LIS1 versus dynactin: P = 0.09, dynein versus NDEL1: P = 0.18, dynein versus LIS1: P = 0.81, NDEL1 versus LIS1: P = 0.13. Kruskal–Wallis test, Dunn post hoc test. **(G)** Instantaneous anterograde velocities of dynein, dynactin, LIS1 and NDEL1 particles in 21–23 DPI neurons. **(H)** The average speed of anterograde dynein and dynactin particles in dual labelled 21–23 DPI neurons. (dynein: 413 tracks, 25 videos, $N$ = 3; dynactin: 732 tracks, 25 videos, $N$ = 3; Co-loc: 57 tracks, 25 videos, $N$ = 3; dynein versus dynactin: ***P = 0.000056, dynein versus co-loc: *P = 0.042, dynactin versus co-loc: ***P = 0.00034. One-way ANOVA, Tukey's post hoc test.) **(I)** The percent colocalization between dynein and dynactin tracks. Boxplots show median, first, and third quartiles. Upper/lower whiskers extend to 1.5× the interquartile range.

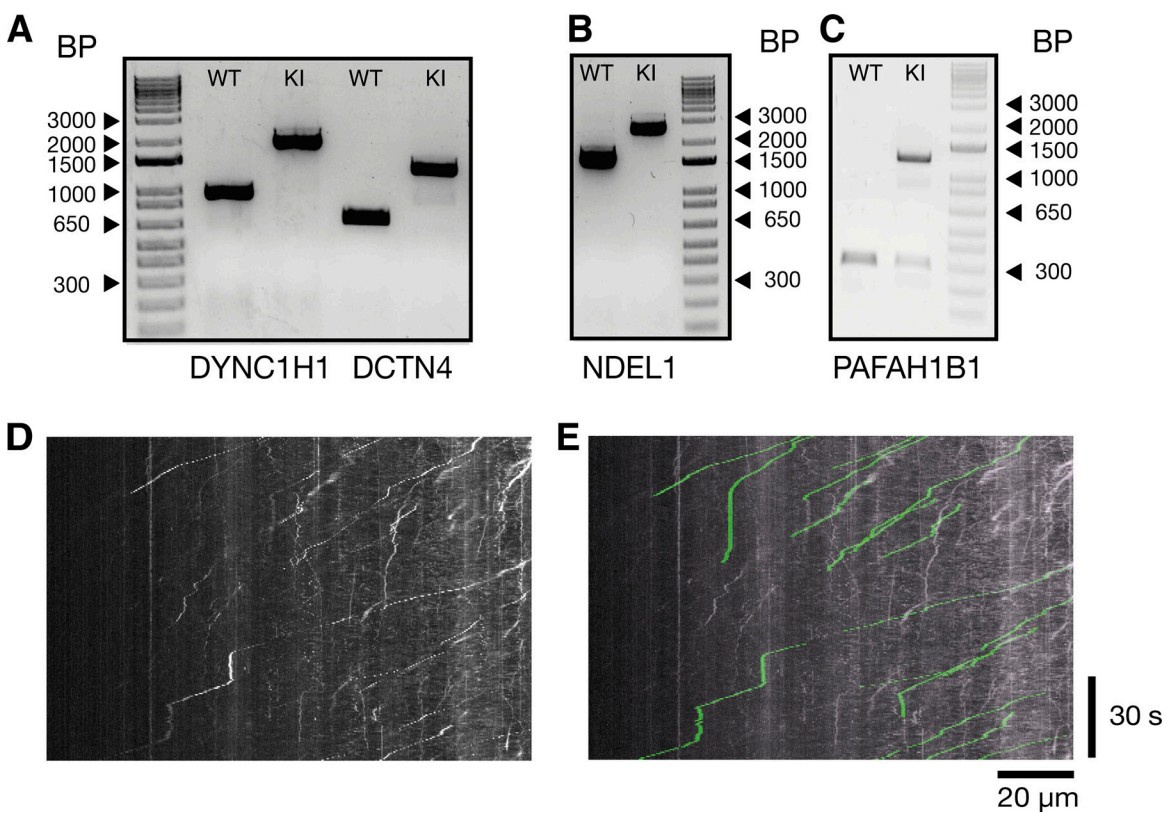

Figure S3.  **Genotyping results of knock-in lines and example analysis. (A)** DNA gel of control cell lines and homozygous knock-in of the Halotag to the *DYNC1H1* and *DCTN4* locus. **(B)** DNA gel of control cell lines and homozygous knock-in of the Halotag to the *NDEL1* locus. **(C)** DNA gel of control cell lines and heterozygous knock-in of the Halotag to the *PAFAH1B1* (LIS1) locus. Comparison between kymograph and spots tracked with Trackmate. **(D)** Kymograph from Fig. 3 A, retrograde dynein tracks (Halo-*DYNC1H1*). **(E)** Tracks extracted from Trackmate have been overlaid on the original kymograph from Fig. 3 A in green. Source data are available for this figure: SourceData FS3.

Video 1.  **Imaging of Halo-DYNC1H1 DPI 21–23 iNeurons treated with 1 nM JFX 650 in the axonal compartment.** Imaged at 30 fps. Playback 60 fps.

Video 2.  **Photobleaching in Halo-*DYNC1H1* DPI 21–23 iNeurons in the axonal compartment treated with 1 nM JFX 554 and 0.5 μM NEM.** Imaged at 30 fps. Playback 60 fps.

Video 3.  **Imaging of Halo-*DYNC1H1* retrograde movement in DPI 21–23 iNeurons treated with 200 nM JFX 554.** Imaged at 2 fps. Playback 20 fps. Includes tracking from Trackmate Fiji plugin.

Video 4.  **Imaging of Halo-ACTR10 retrograde movement in DPI 21–23 iNeurons treated with 200 nM JFX 554.** Imaged at 2 fps. Playback 20 fps.

Video 5.  **Imaging of Halo-*PAFAH1B1* retrograde movement in DPI 21–23 iNeurons treated with 200 nM JFX 554.** Imaged at 2 fps. Playback 20 fps.

Video 6.  **Imaging of Halo-*NDEL1* retrograde movement in DPI 21–23 iNeurons treated with 200 nM JFX 554.** Imaged at 2 fps. Playback 20 fps.

Video 7.  **Imaging of Halo-*PAFAH1B1* anterograde movement in DPI 21–23 iNeurons treated with 200 nM JFX 554.** Imaged at 2 fps. Playback 20 fps.

Video 8.  **Imaging of Halo-*NDEL1* anterograde movement in DPI 21–23 iNeurons treated with 200 nM JFX 554.** Imaged at 2 fps. Playback 20 fps.

Video 9.  **Imaging of dual labeled Halo-*DYNC1H1* and *DCTN4*-SNAP 21-23 DPI iNeurons.** Treated with 200 nM JFX 554 and 1 μM SiR-SNAP. Imaged at 4 fps. Images have been denoised with N2V. Playback 20 fps.

Video 10.  **Example of Noise2Void denoising on *DCTN4*-SNAP 21–23 DPI iNeurons treated with 1 μM SiR-SNAP.** Original video (top) and denoised video (bottom). Imaged at 4 fps. Playback 20 fps.

