## [Peer Review File · The Journal of Cell Biology]

Dynein and dynactin move long-range but are delivered separately to the axon tip

Alexander Fellows, Michaela Bruntraeger, Thomas Burgold, Andrew Bassett, and Andrew Carter

Corresponding Author(s): Andrew Carter, MRC Laboratory of Molecular Biology

Review Timeline:

Submission Date:	2023-09-20
Editorial Decision:	2023-09-27
Revision Received:	2024-01-17
Editorial Decision:	2024-01-29
Revision Received:	2024-01-31

Monitoring Editor: Cassandra Ori-McKenney

Scientific Editor: Dan Simon

Transaction Report:

DOI: <https://doi.org/10.1083/jcb.202309084>

Revision 0

Review #1

1. Evidence, reproducibility and clarity:

Evidence, reproducibility and clarity (Required)

To image dynein in the axon at a single-molecule level, Fellows et al. used neuron-inducible human stem cell lines to Halo/SNAP tag endogenous dynein components by gene editing, and visualized fluorescently labeled protein molecules in differentiated neurons in microfluidic chambers by HILO microscopy-based live imaging. Using those cutting edge technologies, the authors demonstrate that in the axon, not only dynein and dynactin but also the dynein regulators LIS1 and NDEL1 can move long distance retrogradely towards the somatodendritic compartment. They also show that dynein /LIS1 move faster than dynactin/NDEL1 in the anterograde direction, suggesting that they are delivered separately to the distal end of the axon. The approach to study subcellular motility of endogenous dynein/dynactin is creative, the data are solid. I would like to suggest one experiment to support more strongly the authors' conclusions:

If doable, image dynein and dynactin simultaneously in the Halo-DYNC1H1/DCTN4-SNAP iNeurons. Comovement of dynein and dynactin towards the somatodendritic compartment and their separate movement in the anterograde direction along the axon would provide the most convincing evidence for the key claims of the manuscript.

****Referee Cross-Commenting****

I agree with Reviewer 2 that the authors should clarify whether the knockin lines for dynein are homozygous. I also agree with both Reviewers 2 and 3 that the authors should do more analysis of the kymographs to obtain more information.

2. Significance:

Significance (Required)

This is an elegant study on dynein motility and transport in vivo. The experimental approaches and findings presented in this manuscript are very valuable contributions to the field of dynein/dynactin and axonal transport. The results showing that dynein/dynactin can move long-range retrogradely in the axon are in good agreement with previous findings that dynein-driven cargo transport is highly processive, and the data suggesting that dynein and dynactin/NDEL1 are trafficked separately to the distal tip of the axon provide new insights into the regulatory mechanisms for the subcellular distribution and activity of molecular motors. Together these findings provide conceptual advances for understanding axonal transport. They will be of great

interest to not only scientists in the field of intracellular transport but also those in cellular neurobiology.

3. How much time do you estimate the authors will need to complete the suggested revisions:

Estimated time to Complete Revisions (Required)

(Decision Recommendation)

Between 1 and 3 months

Yes

Review #2

1. Evidence, reproducibility and clarity:

Evidence, reproducibility and clarity (Required)

****Summary**** - The authors use a CRISPR knock-in gene editing strategy to label endogenous dynein, dynactin (p62 or Arp11) and dynein regulators (Ndel1 and Lis1) with Halo or SNAP tags. They do this in human iPSC and ESC cell lines engineered to express doxycycline-inducible NGN2 cloned into a "safe harbor" site of the genome. They induce the cells to differentiate into iNeurons using doxycycline and image the tagged proteins in axons with single molecule sensitivity using HILO illumination. The paper is clearly written, the description of the methods is thorough, and the data and figures (including the videos) are of good quality. The use of gene editing to knock the tags into the endogenous gene loci is a superior strategy to classic overexpression strategies. The authors also make effective use of microfluidic chambers to ensure the axons are uniformly orientated and coaligned over a distance of 500µm.

****Major comment**** (requires additional experimentation)

1. While the data presented do certainly suggest that dynein and Lis1 are transported anterogradely on separate vesicular cargoes from dynactin and Ndel1, the study would be much stronger if supported by dual imaging of dynein and dynactin to prove that these proteins do indeed move in association with separate vesicular populations. I would like to see dual-color kymograph traces showing that the proteins move independently. The authors should be able to accomplish this using their dual Halo-DYNC1H1/DCTN4-SNAP hESC line. To acquire and analyze this data might take several months, but it would greatly strengthen this paper. If the authors do this experiment, they may also be able to address the mechanism of reversal of anterograde cargoes which they speculate about in the Discussion, which would add even more interest and insight.

****Minor comments**** (addressable without additional experimentation)

1. The authors deduce that 1-4 Halo fluorochromes corresponds to 1-2 dynein molecules. This implies that the cells are homozygous for the Halo tag, but I do not see this addressed explicitly. The authors should state explicitly whether the lines generated for their study are heterozygous or homozygous for the tag. If the cells are heterozygous, which would seem most likely, then they may be underestimating the number of dyneins per spot and should take this into account.

2. Why are the moving spots lower in intensity than the NEM-treated static spots. It appears to suggest that they may be associated with different structures. This should be clarified and discussed.

3. The authors state in the Results that most of the dynein spots were diffusing, often along microtubules, but they do not visualize microtubules so how do they know this? They may need to remove the phrase "often along microtubules".

4. At the end of the Introduction the authors state that their data "allow us to understand how the dynein machinery drives long-range transport in the axon". This is an overstatement. The "how" in this sentence is not addressed in this study.

5. The conclusion that dynein binds to cargoes stably throughout their transport along the axon is based on measurements of the fastest moving cargoes but the authors do not provide data on the distribution of velocities for the entire population of retrograde cargoes. It is not valid to extrapolate the behavior of a small number of cargoes to the entire population. The average may be much slower than the fastest cargoes. Moreover, even for the fastest organelles the authors cannot say that the dynein is stably bound because they did not track single cargoes and thus do not know that the cargoes moved continuously in one single bout of movement for 500 μm ; it is possible that the cargoes moved in multiple consecutive bouts interrupted by brief pauses and dynein motors may have exchanged between bouts.

6. The authors say that "it is clear that at least some dyneins remain on cargoes throughout their transport along the axon". As explained above, the data do not prove this so this statement should be removed.

7. The authors note that most of the dynein spots were not moving processively and state that this is consistent with prior studies showing that only a subset of dynein is actively involved in transport. However, as they note elsewhere, dynein is both motor and cargo and most axonal dynein is transported at slow average velocities so maybe they should be more explicit about

what they mean by "involved in transport".

8. When the authors note that most of the dynein in axons is transported in the slow component of axonal transport, they should also cite the work of Pfister and colleagues who were the first to show this (PMID 8824315 and 8552592).

9. The authors propose that dynein and Lis1 are transported together but there were significantly fewer anterogradely transported Lis1 particles than dynein particles. This should be discussed.

10. For the statistical analysis, the authors should provide p values in the legends for the comparisons that are judged to be "not significant". The authors should also be consistent in how they label differences that are not significant - they mark them as "ns" in Fig. 1, but in the other figures they do not, leaving some ambiguity about whether particular comparisons were not tested or were found to be not significant. For example, in Fig. 4C the average speed of the dynactin is about 0.5 $\mu\text{m/s}$ greater than for the other proteins and the spread in the data suggest that this could be significant, but no significance is indicated on the plot, implying $p > 0.05$. It is not clear how confident we can be that there is no difference.

****Referee Cross-Commenting****

There seems to be agreement among all three reviewers that the authors should perform dual imaging of dynein and dynactin to prove that these proteins do indeed move together in the retrograde direction but separately in the anterograde direction. This would strengthen the study greatly.

2. Significance:

Significance (Required)

General assessment - There are now multiple papers that have analyzed axonal transport of cargoes in iPSC-derived neurons, but this one appears to be the first to do it by tagging dynein motors and with single-molecule sensitivity. The principal conclusions are (1) that dynein is capable of long-range movement in axons and (2) that dynein moves dynein/Lis1 complexes are transported anterogradely in association with distinct cargoes from dynactin/Ndel1 complexes. The former is a modest conclusion and is entirely expected so not very impactful, but the latter is interesting and novel. The difference between the average velocities for the four proteins in the anterograde and retrograde directions is striking. All four move at similar velocities in the retrograde direction but in the anterograde direction, dynein and Lis1 move significantly faster than dynactin and Ndel1. Given these data, it is reasonable to infer that these proteins are being transported in two separate sets of cargoes. As the authors note in their Discussion, this is important because it could provide a mechanism for transporting dynein components anterogradely in a less active form that could then be activated when the components come together in the distal axon. However, I feel that one critical experiment is missing, which is to perform dual labeling of anterogradely transported dynein and dynactin in the same cells (see major comment). Without this experiment, the evidence is indirect.

Audience - If confirmed by the dual labeling experiment, the authors' conclusions would represent a conceptual and mechanistic insight into the mechanism of bidirectional transport in axons that would be of broad interest to neuronal cell biologists studying neuronal trafficking.

Expertise - This reviewer has expertise in the neuronal cytoskeleton, live imaging and axonal transport and has some experience working with iPSC-derived neurons.

3. How much time do you estimate the authors will need to complete the suggested revisions:

Estimated time to Complete Revisions (Required)

(Decision Recommendation)

Between 1 and 3 months

No

Review #3

1. Evidence, reproducibility and clarity:

Evidence, reproducibility and clarity (Required)

Fellows and coauthors present a single-molecule study toward dynein regulation in axons. They observe that dynein in vivo makes very long runs and that regulators LIS1 and NDEL1 cotransport with dynein all the (retrograde way). Remarkably, different components of the dynein complex appear to be transported in different ways/velocities in the anterograde direction. Overall experiments are well conducted, I only have a couple of important questions regarding data analysis. Some aspects should be explained better, more steps should be shown and here and there I think the authors could, with minimal effort, obtain much more out of their data (see below). Nevertheless, I think this is an important study, one of the first single-molecule efforts to understand axonal transport in the cell (see below). Key findings are important for our

understanding of dynein regulation.

****My concerns:****

- if I look at the kymographs, trajectories appear rather complex, pausing, standing still, moving and everything mixed. The explanation of how actual trajectories are extracted and on what basis is very short, too short for me. I think the authors should expand this. Furthermore, I think it would be good if the authors would present, in their kymographs examples of the tracked (and also the not included) tracks. Maybe in supplementary info.
- I found 'velocity' ill defined. I get the impression, judging from the number of points (compared to the other parameters) that the authors determine the average velocity of each individual trajectory. That is an important parameter (but should indeed be called 'trajectory averaged' velocity), but might not be the only one useful to learn from the data, where trajectories do not always appear to have constant speeds (pausing, etc.). Why do the authors not determine point-to-point velocities and plot histograms of those for all the trajectories (simply plot histograms of all the displacements between subsequent data points in trajectories)? This might provide great insight into the actual maximum velocity and the fraction of pausing or moving in opposite direction etc., providing much more molecular detail than currently extracted from the data.
- I was a bit surprised to read that the authors have gone to the effort to create a dual-color labeled cell line, but did not do actual correlative two-color measurements (or at least show them). It would be so insightful to see dynein and dynactin move separately in the anterograde direction.

****Referee Cross-Commenting****

I think we agree on the key points:

- in principle, great study
- quantification / tracking could go a bit further and should be explained better
- manuscript / conclusions would be strengthened substantially if the authors could do some 2-color experiments to correlated dynein / dynactin movements in anterograde vs retrograde direction.

2. Significance:

Significance (Required)

I think this is an important and exciting manuscript. As an in vivo single-molecule biophysicist with great interest in intracellular transport, I have been astonished in the lack of people trying to take single-molecule data on the motor involved, in particular neurons. I believe this is the only way to find out how transport actually works and what role motors play. Mutants is not enough, bulk data is not enough, in vitro is not enough. This is what the field needs (and many in the field do not seem to be aware of this...). Great that Fellows and coauthors took on this task and show some really exciting data. I am not an expert on their stem-cell labeling approach so cannot judge on that. The imaging seems to be done well. As discussed above, I think there might be much more in the data than the authors now get out, so I would encourage them to do some additional

analysis. But overall, this effort is important and I think the conclusions will stand and provide important new insights in dynein regulation in the cell.

3. How much time do you estimate the authors will need to complete the suggested revisions:

Estimated time to Complete Revisions (Required)

(Decision Recommendation)

Between 1 and 3 months

Yes

Revision Plan

Manuscript number: RC-2023-02089

Corresponding author(s): Andrew, Carter

1. General Statements

We thank the reviewers for their positive comments and also for their thorough evaluation and constructive feedback. Please see the cover letter for statements about goals and highlights of our study.

2. Description of the planned revisions

- Two color imaging of Halo-DYNC1H1 and DCTN4-SNAP movements in iNeurons

We agree that two color imaging to show that dynein and dynactin move separately anterogradely, but together in the retrograde direction, would be the ultimate proof of our model. However, the reason we did not initially include this is because imaging the SNAP-SiR ligand in the retrograde direction is difficult due to non-specific labelling. In other words when we labelled untagged cell lines with SNAP-SiR in the synaptic compartment we see lots of retrograde movement in the axon. This “background” was not observed with Halo-JFX544/650 ligands and was not seen for anterograde movement for SNAP-SiR or Halotag ligands (i.e. when labelling was performed in the somatic compartment).

Our plan is to perform new two-color studies on anterograde movement. These will show if the fast dynactin moves together with dynein and provide visualization of the different speeds of dynein and dynactin in the same axon. We will also attempt to improve the specificity of retrograde SNAP labelling, by testing a wider range of SNAP-fluorophores. If we are able to avoid the non-specific labelling, we will include these data.

- Reviewers suggest further analysis of tracking data including histogram of displacement between data points

We are in the process of performing this analysis and will include it in the final revision.

3. Description of the revisions that have already been incorporated in the transferred manuscript

Reviewer #1

1. If doable, image dynein and dynactin simultaneously in the Halo-DYNC1H1/DCTN4-SNAP iNeurons. Co-movement of dynein and dynactin towards the somatodendritic compartment and their separate movement in the anterograde direction along the axon would provide the most convincing evidence for the key claims of the manuscript.

Please see the planned revision section for our response

Revision Plan

Reviewer #2

Major comment (requires additional experimentation)

1. While the data presented do certainly suggest that dynein and Lis1 are transported anterogradely on separate vesicular cargoes from dynactin and Ndel1, the study would be much stronger if supported by dual imaging of dynein and dynactin to prove that these proteins do indeed move in association with separate vesicular populations. I would like to see dual-color kymograph traces showing that the proteins move independently. The authors should be able to accomplish this using their dual Halo-DYNC1H1/DCTN4-SNAP hESC line. To acquire and analyze this data might take several months, but it would greatly strengthen this paper. If the authors do this experiment, they may also be able to address the mechanism of reversal of anterograde cargoes which they speculate about in the Discussion, which would add even more interest and insight.

Please see the planned revision section for our response

Minor comments (addressable without additional experimentation)

1. The authors deduce that 1-4 Halo fluorochromes corresponds to 1-2 dynein molecules. This implies that the cells are homozygous for the Halo tag, but I do not see this addressed explicitly. The authors should state explicitly whether the lines generated for their study are heterozygous or homozygous for the tag. If the cells are heterozygous, which would seem most likely, then they may be underestimating the number of dyneins per spot and should take this into account.

We have added whether lines are homozygous or heterozygous to the manuscript. We also include a new Supplementary Figure panel (Fig S6) showing the genotyping data. In summary, all lines are homozygous except for PFAH1B1-Halo (hESCs) which is heterozygous.

2. Why are the moving spots lower in intensity than the NEM-treated static spots. It appears to suggest that they may be associated with different structures. This should be clarified and discussed.

Our data suggest that the fast-moving spots have fewer dyneins than NEM treated static spots. We suggest this is because the fast-moving cargoes are smaller than the average cargo and therefore have fewer dyneins on them. This is also supported by the smaller number of dyneins reported previously on endosomes as compared to the large lysosomes. We have clarified this in the discussion (page 7-8).

3. The authors state in the Results that most of the dynein spots were diffusing, often along microtubules, but they do not visualize microtubules so how do they know this? They may need to remove the phrase "often along microtubules".

This has been removed.

4. At the end of the Introduction the authors state that their data "allow us to understand how the dynein machinery drives long-range transport in the axon". This is an overstatement. The "how" in this sentence is not addressed in this study.

We have softened the sentence by adding the phrase "better understand".

Revision Plan

5. The conclusion that dynein binds to cargos stably throughout their transport along the axon is based on measurements of the fastest moving cargos but the authors do not provide data on the distribution of velocities for the entire population of retrograde cargos. It is not valid to extrapolate the behavior of a small number of cargos to the entire population. The average may be much slower than the fastest cargos. Moreover, even for the fastest organelles the authors cannot say that the dynein is stably bound because they did not track single cargos and thus do not know that the cargos moved continuously in one single bout of movement for 500 μm ; it is possible that the cargos moved in multiple consecutive bouts interrupted by brief pauses and dynein motors may have exchanged between bouts.

We have added a section to the discussion to highlight that other cargos may behave differently from the fastest ones (page 7). We have also clarified the assumptions that lead us to expect a slower arrival time of the first signal (page 5).

6. The authors say that "it is clear that at least some dyneins remain on cargos throughout their transport along the axon". As explained above, the data do not prove this so this statement should be removed.

We have softened this sentence from "it is clear" to "our results suggest" and explained in more detail why we make this conclusion

7. The authors note that most of the dynein spots were not moving processively and state that this is consistent with prior studies showing that only a subset of dynein is actively involved in transport. However, as they note elsewhere, dynein is both motor and cargo and most axonal dynein is transported at slow average velocities so maybe they should be more explicit about what they mean by "involved in transport".

We have clarified we mean fast axonal transport and thank the reviewer for highlighting this point.

8. When the authors note that most of the dynein in axons is transported in the slow component of axonal transport, they should also cite the work of Pfister and colleagues who were the first to show this (PMID 8824315 and 8552592).

This was an omission on our part. The references have now been added.

9. The authors propose that dynein and Lis1 are transported together but there were significantly fewer anterogradely transported Lis1 particles than dynein particles. This should be discussed.

We have added more information to the discussion. Although we cannot rule out this effect being due to the heterozygous tagging of our LIS1 cell line, we do not witness the same decrease in events in the retrograde direction. Therefore, we believe there is a subset of anterogradely moving dynein lacking LIS1. As discussed in the manuscript, this subset may already be bound to dynactin and therefore not require LIS1.

10. For the statistical analysis, the authors should provide p values in the legends for the comparisons that are judged to be "not significant". The authors should also be consistent in how they label differences that are not significant - they mark them as "ns" in Fig. 1, but in the other figures they do not, leaving some ambiguity about whether particular comparisons were not tested or were found to be not significant.

Revision Plan

For example, in Fig. 4C the average speed of the dynactin is about 0.5 $\mu\text{m/s}$ greater than for the other proteins and the spread in the data suggest that this could be significant, but no significance is indicated on the plot, implying $p > 0.05$. It is not clear how confident we can be that there is no difference.

We have now included all p values in the figure legends and have removed the “ns” in Fig 1D. In our revised manuscript, only significant differences are highlighted in the figures.

Reviewer #3

- if I look at the kymographs, trajectories appear rather complex, pausing, standing still, moving and everything mixed. The explanation of how actual trajectories are extracted and on what basis is very short, too short for me. I think the authors should expand this. Furthermore, I think it would be good if the authors would present, in their kymographs examples of the tracked (and also the not included) tracks. Maybe in supplementary info.

The analysis of this data used the Trackmate Fiji plugin. This tracks spots frame to frame in a movie and then outputs the data of the tracks. No data was extracted from kymographs but they were used as a graphical illustration of the moving spots. To better explain our analysis pipeline, we have expanded our methods section and have added an example of a tracked movie (Video 15) as well as highlighted the tracked spots in one kymograph example (Figure 7S).

- I found 'velocity' ill defined. I get the impression, judging from the number of points (compared to the other parameters) that the authors determine the average velocity of each individual trajectory. That is an important parameter (but should indeed be called 'trajectory averaged' velocity), but might not be the only one useful to learn from the data, where trajectories do not always appear to have constant speeds (pausing, etc.). Why do the authors not determine point-to-point velocities and plot histograms of those for all the trajectories (simply plot histograms of all the displacements between subsequent data points in trajectories)? This might provide great insight into the actual maximum velocity and the fraction of pausing or moving in opposite direction etc., providing much more molecular detail than currently extracted from the data.

The reviewer is correct. We have measured the average velocity of the spots from the beginning of the track to the end. We have clarified this in the text. Furthermore, as stated above in the revision plan, we are currently doing the additional analysis and will include it in the final revision

- I was a bit surprised to read that the authors have gone to the effort to create a dual-color labeled cell line, but did not do actual correlative two-color measurements (or at least show them). It would be so insightful to see dynein and dynactin move separately in the anterograde direction.

Please see the planned revision section for our response

September 27, 2023

Re: JCB manuscript #202309084T

Dr. Andrew Carter
MRC Laboratory of Molecular Biology
Francis Crick Ave
Cambridge CB2 0QH

Dear Dr. Carter,

Thank you for submitting your manuscript entitled "Dynein and dynactin move long-range but are delivered separately to the axon tip." We have assessed the manuscript, the reports from Review Commons, and your revision plan. We agree that the study is suitable for JCB and invite you to submit a revision as outlined in the revision plan. We think the work would be most appropriate for our Report format and ask that you format the revision according to the specific formatting guidelines below.

GENERAL GUIDELINES:

Text limits: Character count for a Report is < 20,000, not including spaces. Count includes title page, abstract, introduction, the joint Results & Discussion, and acknowledgments. Count does not include materials and methods, figure legends, references, tables, or supplemental legends.

Figures: Reports may have up to 5 main text figures. Figures can span up to a full page so we believe you should be able to combine the current and future data in order to fit within this limit. To avoid delays in production, figures must be prepared according to the policies outlined in our Instructions to Authors, under Data Presentation, <https://jcb.rupress.org/site/misc/ifora.xhtml>. All figures in accepted manuscripts will be screened prior to publication.

Supplemental information: Reports generally have up to 3 supplemental figures, which can also span a full page. In this case, we will be able to give you extra space if needed but please try to consolidate the data as much as possible. A summary of all supplemental material should appear at the end of the Materials and methods section.

*****IMPORTANT:** It is JCB policy that if requested, original data images must be made available. Failure to provide original images upon request will result in unavoidable delays in publication. Please ensure that you have access to all original microscopy and blot data images before submitting your revision. ***

Full guidelines are also available on our Instructions for Authors page, <https://jcb.rupress.org/submission-guidelines#revised>

Please note that JCB now requires authors to submit Source Data used to generate figures containing gels and Western blots with all revised manuscripts. This Source Data consists of fully uncropped and unprocessed images for each gel/blot displayed in the main and supplemental figures. Since your paper includes cropped gel and/or blot images, please be sure to provide one Source Data file for each figure that contains gels and/or blots along with your revised manuscript files. File names for Source Data figures should be alphanumeric without any spaces or special characters (i.e., SourceDataF#, where F# refers to the associated main figure number or SourceDataFS# for those associated with Supplementary figures). The lanes of the gels/blots should be labeled as they are in the associated figure, the place where cropping was applied should be marked (with a box), and molecular weight/size standards should be labeled wherever possible. Source Data files will be made available to reviewers during evaluation of revised manuscripts and, if your paper is eventually published in JCB, the files will be directly linked to specific figures in the published article.

The typical timeframe for revisions is three to four months. While most universities and institutes have reopened labs and allowed researchers to begin working at nearly pre-pandemic levels, we at JCB realize that the lingering effects of the COVID-19 pandemic may still be impacting some aspects of your work, including the acquisition of equipment and reagents. Therefore, if you anticipate any difficulties in meeting this aforementioned revision time limit, please contact us and we can work with you to find an appropriate time frame for resubmission. Please note that papers are generally considered through only one revision cycle, so any revised manuscript will likely be either accepted or rejected.

Thank you for this interesting contribution to Journal of Cell Biology. You can contact us at the journal office with any questions, cellbio@rockefeller.edu.

Sincerely,

Kassandra Ori-McKenney, PhD
Monitoring Editor
Journal of Cell Biology

Dan Simon, PhD
Scientific Editor
Journal of Cell Biology

Dear Dr Ori-Mckenney and Dr Simon,

Thank you for your interest in our manuscript. We have addressed all the reviewer's comments and provide a detailed response below. In response to comments from all three reviewers we performed additional experiments and analysis and added these to the manuscript (Additional Experiments and Analysis - below). Any text changes that have occurred due to these have been highlighted. Furthermore, we have shortened our manuscript to fit into the report style of JCB.

Additional Experiments and Analysis:

- The reviewers asked us to include two colour imaging of Halo-*DYNC1H1* and *DCTN4*-SNAP movements in iNeurons to test our hypothesis that dynein and dynactin are predominantly transported separately in the anterograde direction.

We performed two colour imaging with Halo-*DYNC1H1* and *DCTN4*-SNAP in the anterograde direction. This data has been added to the manuscript (Fig 5 and S2, text lines 183-189). In agreement with the different average anterograde speeds, we found that 88-90% of dynein and dynactin show no colocalization. Of the remaining 10-12% of particles which showed the presence of both dynein and dynactin the average speed was $\sim 3.08 \mu\text{m/s}$. This agrees with our suggestion that faster moving dynactin molecules were present together with dynein. In other words, our data support the idea that while the majority of dyneins and dynactins are moved to the tip separately, there are a subset of cargos which have both present.

We agreed that seeing both dynein and dynactin moving together in the retrograde direction would have been strengthened our model, however, due to non-specific labelling we were unable to analyse their movement in the retrograde direction. In other words when we labelled untagged cell lines with SNAP-SiR in the synaptic compartment we see lots of retrograde movement in the axon. This "background" was not observed with Halo-JFX544/650 ligands and was not seen for anterograde movement for SNAP-SiR or Halotag ligands (i.e. when labelling was performed in the somatic compartment). We tested multiple SNAP dyes and various concentration but could never eliminate the non-specific signal (see figure below).

- Reviewers suggest further analysis of tracking data including histogram of displacement between data points

We further analysed our data by investigating how the instantaneous velocity differed between the components of the dynein machinery (Fig S2). In agreement with our average speed analysis, we saw that in retrograde direction components of the dynein machinery moved in a similar fashion, with the majority of velocities being between 1.8- 2.2 $\mu\text{m/s}$. When analysing instantaneous velocities in the anterograde direction we again saw the striking difference between dynein/LIS1 and dynactin/NDEL1. This data further supports our hypothesis that core dynein machinery is kept apart during transit towards the distal tip. Furthermore, our new analysis highlights how uni-directional these proteins are when moving in the axon. We observed very few, if any, anterograde movements when analysing retrograde transport and the same was true when visualising anterograde transport.

Other reviewer comments

We addressed all the other reviewers' comments in our initial submission to JCB, however we have listed these again below for completion.

Reviewer#2

Minor comments (addressable without additional experimentation)

1. The authors deduce that 1-4 Halo fluorochromes corresponds to 1-2 dynein molecules. This implies that the cells are homozygous for the Halo tag, but I do not see this addressed explicitly. The authors should state explicitly whether the lines generated for their study are heterozygous or homozygous for the tag. If the cells are heterozygous, which would seem most likely, then they may be underestimating the number of dyneins per spot and should take this into account.

We have added whether lines are homozygous or heterozygous to the manuscript. We also include a new Supplementary Figure panel (Fig S6) showing the genotyping data. In summary, all lines are homozygous except for PAFAH1B1-Halo (hESCs) which is heterozygous.

2. Why are the moving spots lower in intensity than the NEM-treated static spots. It appears to suggest that they may be associated with different structures. This should be clarified and discussed.

Our data suggest that the fast-moving spots have fewer dyneins than NEM treated static spots. We suggest this is because the fast-moving cargos are smaller than the average cargo and therefore have fewer dyneins on them. This is also supported by the smaller number of dyneins reported previously on endosomes as compared to the large lysosomes. We have clarified this in the discussion (page 7-8).

3. The authors state in the Results that most of the dynein spots were diffusing, often along microtubules, but they do not visualize microtubules so how do they know this? They may need to remove the phrase "often along microtubules".

This has been removed.

4. At the end of the Introduction the authors state that their data "allow us to understand how the dynein machinery drives long-range transport in the axon". This is an overstatement. The "how" in this sentence is not addressed in this study.

We have softened the sentence by adding the phrase “better understand”.

5. The conclusion that dynein binds to cargos stably throughout their transport along the axon is based on measurements of the fastest moving cargoes but the authors do not provide data on the distribution of velocities for the entire population of retrograde cargoes. It is not valid to extrapolate the behavior of a small number of cargoes to the entire population. The average may be much slower than the fastest cargoes. Moreover, even for the fastest organelles the authors cannot say that the dynein is stably bound because they did not track single cargoes and thus do not know that the cargoes moved continuously in one single bout of movement for 500 μm ; it is possible that the cargoes moved in multiple consecutive bouts interrupted by brief pauses and dynein motors may have exchanged between bouts.

We have added a section to the discussion to highlight that other cargoes may behave differently from the fastest ones (page 7). We have also clarified the assumptions that lead us to expect a slower arrival time of the first signal (page 5).

6. The authors say that “it is clear that at least some dyneins remain on cargoes throughout their transport along the axon”. As explained above, the data do not prove this so this statement should be removed.

We have softened this sentence from “it is clear” to “our results suggest” and explained in more detail why we make this conclusion

7. The authors note that most of the dynein spots were not moving processively and state that this is consistent with prior studies showing that only a subset of dynein is actively involved in transport. However, as they note elsewhere, dynein is both motor and cargo and most axonal dynein is transported at slow average velocities so maybe they should be more explicit about what they mean by “involved in transport”.

We have clarified we mean fast axonal transport and thank the reviewer for highlighting this point.

8. When the authors note that most of the dynein in axons is transported in the slow component of axonal transport, they should also cite the work of Pfister and colleagues who were the first to show this (PMID 8824315 and 8552592).

This was an omission on our part. The references have now been added.

9. The authors propose that dynein and Lis1 are transported together but there were significantly fewer anterogradely transported Lis1 particles than dynein particles. This should be discussed.

We have added more information to the discussion. Although we cannot rule out this effect being due to the heterozygous tagging of our LIS1 cell line, we do not witness the same decrease in events in the retrograde direction. Therefore, we believe there is a subset of anterogradely moving dynein lacking LIS1. As discussed in the manuscript, this subset may already be bound to dynactin and therefore not require LIS1.

10. For the statistical analysis, the authors should provide p values in the legends for the comparisons that are judged to be “not significant”. The authors should also be consistent in how they label differences that are not significant - they mark them as “ns” in Fig. 1, but in the other figures they do

January 17, 2024

not, leaving some ambiguity about whether particular comparisons were not tested or were found to be not significant. For example, in Fig. 4C the average speed of the dynactin is about 0.5 $\mu\text{m/s}$ greater than for the other proteins and the spread in the data suggest that this could be significant, but no significance is indicated on the plot, implying $p > 0.05$. It is not clear how confident we can be that there is no difference.

We have now included all p values in the figure legends and have removed the “ns” in Fig 1D. In our revised manuscript, only significant differences are highlighted in the figures.

Reviewer#3

- if I look at the kymographs, trajectories appear rather complex, pausing, standing still, moving and everything mixed. The explanation of how actual trajectories are extracted and on what basis is very short, too short for me. I think the authors should expand this. Furthermore, I think it would be good if the authors would present, in their kymographs examples of the tracked (and also the not included) tracks. Maybe in supplementary info.

The analysis of this data used the Trackmate Fiji plugin. This tracks spots frame to frame in a movie and then outputs the data of the tracks. No data was extracted from kymographs but they were used as a graphical illustration of the moving spots. To better explain our analysis pipeline, we have expanded our methods section and have added an example of a tracked movie (Video 3) as well as highlighted the tracked spots in one kymograph example (Figure 3S).

-I found 'velocity' ill defined. I get the impression, judging from the number of points (compared to the other parameters) that the authors determine the average velocity of each individual trajectory. That is an important parameter (but should indeed be called 'trajectory averaged' velocity), but might not be the only one useful to learn from the data, where trajectories do not always appear to have constant speeds (pausing, etc.). Why do the authors not determine point-to-point velocities and plot histograms of those for all the trajectories (simply plot histograms of all the displacements between subsequent data points in trajectories)? This might provide great insight into the actual maximum velocity and the fraction of pausing or moving in opposite direction etc., providing much more molecular detail than currently extracted from the data.

The reviewer is correct. We have measured the average velocity of the spots from the beginning of the track to the end. We have clarified this in the text. Furthermore, as stated above in the revision plan, we are currently doing the additional analysis and will include it in the final revision

Yours sincerely,

Dr Andrew Carter
Programme Leader, Structural Studies Division
cartera@mrc-lmb.cam.ac.uk

January 29, 2024

RE: JCB Manuscript #202309084R

Dr. Andrew P Carter
MRC Laboratory of Molecular Biology
Structural Studies
Francis Crick Ave
Cambridge CB2 0QH
United Kingdom

Dear Dr. Carter,

Thank you for submitting your revised manuscript entitled "Dynein and dynactin move long-range but are delivered separately to the axon tip." We would be happy to publish your paper in JCB pending final revisions necessary to meet our formatting guidelines (see details below).

A. MANUSCRIPT ORGANIZATION AND FORMATTING:

1) Text limits: Character count for Reports is < 20,000, not including spaces. Count includes title page, abstract, introduction, results, discussion, and acknowledgments. Count does not include materials and methods, figure legends, references, tables, or supplemental legends.

2) Figure formatting: Reports may have up to 5 main text figures. Scale bars must be present on all microscopy images, including inset magnifications. Molecular weight or nucleic acid size markers must be included on all gel electrophoresis. Please add a scale bar to figure S1E.

Also, please avoid pairing red and green for images and graphs to ensure legibility for color-blind readers. If red and green are paired for images, please ensure that the particular red and green hues used in micrographs are distinctive with any of the colorblind types. If not, please modify colors accordingly or provide separate images of the individual channels.

3) Statistical analysis: Error bars on graphic representations of numerical data must be clearly described in the figure legend. The number of independent data points (n) represented in a graph must be indicated in the legend. Please, indicate whether 'n' refers to technical or biological replicates (i.e. number of analyzed cells, samples or animals, number of independent experiments). If independent experiments with multiple biological replicates have been performed, we recommend using distribution-reproducibility SuperPlots (please see Lord et al., JCB 2020) to better display the distribution of the entire dataset, and report statistics (such as means, error bars, and P values) that address the reproducibility of the findings.

Statistical methods should be explained in full in the materials and methods. For figures presenting pooled data the statistical measure should be defined in the figure legends. Please also be sure to indicate the statistical tests used in each of your experiments (both in the figure legend itself and in a separate methods section) as well as the parameters of the test (for example, if you ran a t-test, please indicate if it was one- or two-sided, etc.). Also, if you used parametric tests, please indicate if the data distribution was tested for normality (and if so, how). If not, you must state something to the effect that "Data distribution was assumed to be normal but this was not formally tested."

4) Materials and methods: Should be comprehensive and not simply reference a previous publication for details on how an experiment was performed. Please provide full descriptions (at least in brief) in the text for readers who may not have access to referenced manuscripts. The text should not refer to methods "...as previously described."

5) For all cell lines, vectors, constructs/cDNAs, etc. - all genetic material: please include database / vendor ID (e.g., Addgene, ATCC, etc.) or if unavailable, please briefly describe their basic genetic features, even if described in other published work or gifted to you by other investigators (and provide references where appropriate). Please be sure to provide the sequences for all of your oligos: primers, si/shRNA, RNAi, gRNAs, etc. in the materials and methods. You must also indicate in the methods the source, species, and catalog numbers/vendor identifiers (where appropriate) for all of your antibodies, including secondary. If antibodies are not commercial, please add a reference citation if possible.

- 6) Microscope image acquisition: The following information must be provided about the acquisition and processing of images:
- Make and model of microscope
 - Type, magnification, and numerical aperture of the objective lenses
 - Temperature
 - Imaging medium
 - Fluorochromes
 - Camera make and model
 - Acquisition software
 - Any software used for image processing subsequent to data acquisition. Please include details and types of operations involved (e.g., type of deconvolution, 3D reconstitutions, surface or volume rendering, gamma adjustments, etc.).
- 7) References: There is no limit to the number of references cited in a manuscript. References should be cited parenthetically in the text by author and year of publication. Abbreviate the names of journals according to PubMed.
- 8) Supplemental materials: Reports may have up to 3 supplemental figures and 10 videos. Please also note that tables, like figures, should be provided as individual, editable files. A summary of all supplemental material should appear at the end of the Materials and methods section. Please include one brief sentence per item.
- 9) Video legends: Should describe what is being shown, the cell type or tissue being viewed (including relevant cell treatments, concentration and duration, or transfection), the imaging method (e.g., time-lapse epifluorescence microscopy), what each color represents, how often frames were collected, the frames/second display rate, and the number of any figure that has related video stills or images.
- 10) eTOC summary: A ~40-50 word summary that describes the context and significance of the findings for a general readership should be included on the title page. The statement should be written in the present tense and refer to the work in the third person. It should begin with "First author name(s) et al..." to match our preferred style.
- 11) Conflict of interest statement: JCB requires inclusion of a statement in the acknowledgements regarding competing financial interests. If no competing financial interests exist, please include the following statement: "The authors declare no competing financial interests." If competing interests are declared, please follow your statement of these competing interests with the following statement: "The authors declare no further competing financial interests."
- 12) A separate author contribution section is required following the Acknowledgments in all research manuscripts. All authors should be mentioned and designated by their first and middle initials and full surnames. We encourage use of the CRediT nomenclature (<https://casrai.org/credit/>).
- 13) ORCID IDs: ORCID IDs are unique identifiers allowing researchers to create a record of their various scholarly contributions in a single place. Please note that ORCID IDs are required for all authors. At resubmission of your final files, please be sure to provide your ORCID ID and those of all co-authors.
- 14) JCB requires authors to submit Source Data used to generate figures containing gels and Western blots with all revised manuscripts. This Source Data consists of fully uncropped and unprocessed images for each gel/blot displayed in the main and supplemental figures. Since your paper includes cropped gel and/or blot images, please be sure to provide one Source Data file for each figure that contains gels and/or blots along with your revised manuscript files. File names for Source Data figures should be alphanumeric without any spaces or special characters (i.e., SourceDataF#, where F# refers to the associated main figure number or SourceDataFS# for those associated with Supplementary figures). The lanes of the gels/blots should be labeled as they are in the associated figure, the place where cropping was applied should be marked (with a box), and molecular weight/size standards should be labeled wherever possible. Source Data files will be directly linked to specific figures in the published article. Source Data Figures should be provided as individual PDF files (one file per figure). Authors should endeavor to retain a minimum resolution of 300 dpi or pixels per inch. Please review our instructions for export from Photoshop, Illustrator, and PowerPoint here: <https://rupress.org/jcb/pages/submission-guidelines#revised>
- 15) Journal of Cell Biology now requires a data availability statement for all research article submissions. These statements will be published in the article directly above the Acknowledgments. The statement should address all data underlying the research presented in the manuscript. Please visit the JCB instructions for authors for guidelines and examples of statements at (<https://rupress.org/jcb/pages/editorial-policies#data-availability-statement>).

B. FINAL FILES:

Thank you for this interesting contribution, we look forward to publishing your paper in Journal of Cell Biology.

Sincerely,

Kassandra Ori-McKenney, PhD
Monitoring Editor
Journal of Cell Biology

Dan Simon, PhD
Scientific Editor
Journal of Cell Biology